# Nonlinear optical diode effect in a magnetic Weyl semimetal

Christian Tzschaschel [1,10] ✉, Jian-Xiang Qiu [1], Xue-Jian Gao [2], Hou-Chen Li[1], Chunyu Guo [3,4], Hung-Yu Yang [5], Cheng-Ping Zhang [2], Ying-Ming Xie [2], Yu-Fei Liu[1], Anyuan Gao [1], Damien Bérubé[1], Thao Dinh[1], Sheng-Chin Ho[1], Yuqiang Fang[6,7], Fuqiang Huang [6,7], Johanna Nordlander [8], Qiong Ma [5,9], Fazel Tafti [5], Philip J. W. Moll [3,4], Kam Tuen Law [2] & Su-Yang Xu [1] ✉

Diode effects are of great interest for both fundamental physics and modern technologies. Electrical diode effects (nonreciprocal transport) have been observed in Weyl systems. Optical diode effects arising from the Weyl fermions have been theoretically considered but not probed experimentally. Here, we report the observation of a nonlinear optical diode effect (NODE) in the magnetic Weyl semimetal CeAlSi, where the magnetization introduces a pronounced directionality in the nonlinear optical second-harmonic generation (SHG). We demonstrate a six-fold change of the measured SHG intensity between opposite propagation directions over a bandwidth exceeding 250 meV. Supported by density-functional theory, we establish the linearly dispersive bands emerging from Weyl nodes as the origin of this broadband effect. We further demonstrate current-induced magnetization switching and thus electrical control of the NODE. Our results advance ongoing research to identify novel nonlinear optical/transport phenomena in magnetic topological materials and further opens new pathways for the unidirectional manipulation of light.

Diode effects, i.e., phenomena that exhibit a preferred direction, are at the heart of new fundamental physics and modern technologies. A primary example is an electrical diode, which readily conducts current in one direction, but has a high resistance in the opposite direction. In analogy to electrical diodes, optical diodes are characterized by a directionally asymmetric propagation of light. Recent theoretical studies have shown that such an optical diode effect can reveal fundamentally new physics, such as the quantum metric of Bloch wavefunctions[1–4]. Moreover, the optical diode would be a crucial component of various optical technologies including photonics, where light instead of electrical current is envisioned as information carrier. For instance, in the microwave regime, the optical diode effect has been proposed for telecommunications[5,6]. Such prospects have inspired a flurry of experimental studies of optical diode effects (such as the nonreciprocal directional dichroism) especially in magnetoelectric wide-bandgap insulators[7,8]. However, the observed optical diode effect is typically very weak with the difference between forward and backward propagation being less than 1% unless the photon

[1]Department of Chemistry and Chemical Biology, Harvard University, Cambridge, MA 02138, USA. [2]Department of Physics, Hong Kong University of Science and Technology, Clear Water Bay, Hong Kong, China. [3]Max Planck Institute for the Structure and Dynamics of Matter, Hamburg, Germany. [4]Laboratory of Quantum Materials (QMAT), Institute of Materials (IMX), École Polytechnique Fédérale de Lausanne (EPFL), CH-1015 Lausanne, Switzerland. [5]Department of Physics, Boston College, Chestnut Hill, MA, USA. [6]State Key Laboratory of High Performance Ceramics and Superfine Microstructure, Shanghai Institute of Ceramics, Chinese Academy of Science, Shanghai, China. [7]State Key Laboratory of Rare Earth Materials Chemistry and Applications, College of Chemistry and Molecular Engineering Peking University, Beijing, China. [8]Department of Physics, Harvard University, Cambridge, MA 02138, USA. [9]CIFAR Azrieli Global Scholars program, CIFAR, Toronto, Ontario, Canada. [10]Present address: Max-Born Institute for Nonlinear Optics and Short Pulse Spectroscopy, Berlin, Germany. ✉e-mail: tzschaschel@mbi-berlin.de; suyangxu@fas.harvard.edu

energy is tuned to specific narrow electronic or magnetic resonances with bandwidths often below 5 meV.

In addition to its propagation direction, light has another important degree of freedom, the color (frequency). Therefore, it is possible to conceptualize the nonlinear optical diode effect (NODE), where the photon frequency changes during the diode process. A second-order diode effect is a process where the optical second-harmonic generation (SHG) of a material along the forward and backward propagation directions differ. In terms of fundamental physics, the NODE may reveal novel quantum geometrical phenomena distinct from the linear optical diode[4,9–12]. Also, nonlinear optical processes (e.g. SHG) can lead to a much stronger diode effect (i.e., a larger contrast between the forward and backward direction) as SHG supports much stronger crystallographic selection rules[13,14]. In terms of technology, the NODE may also enable novel optical applications such as unidirectional and mode-locked lasers[15] and novel designs for optical isolators for optical communication[16].

We show in Fig. 1a a schematic representation of the NODE. From the point of view of symmetry, we note that the two opposite propagation directions are related by a mirror operation (mirror perpendicular to $\hat{x}$ in Fig. 1a). The observation of a directionally dependent SHG intensity, i.e., a NODE, thus requires the presence of different contributions to the SHG intensity that are even and odd with respect to

the mirror operation. This distinguishes the NODE from other SHG interference phenomena; most notably magnetic SHG contrast[17] and nonreciprocal SHG[13,14]. Magnetic contrast arises due to interference between i-type (time-reversal $\mathcal{T}$ even) and c-type ($\mathcal{T}$ odd) contributions to the SHG response and thus strictly requires $\mathcal{T}$ symmetry to be broken. Nonreciprocal SHG arises from a $\mathbf{k}$-dependent nonlinear optical susceptibility ($\mathbf{k}$ being the photon momentum). Therefore, nonreciprocal SHG contains $\mathbf{k}$-even contributions, such as electric-dipole SHG (which is $\mathcal{P}$ odd) and $\mathbf{k}$-odd contributions, such as as magnetic-dipole or electric-quadrupole SHG (which are $\mathcal{P}$ even)[13,14]. In contrast, the NODE is observable in the absence of $\mathcal{T}$ or $\mathcal{P}$ symmetry breaking or can be realized purely based on electric-dipole SHG. The NODE generalizes the ground-breaking concept of nonreciprocal SHG and may occur concomitantly with a magnetic contrast (see Supplementary Section 1D for an extended discussion of SHG interference phenomena and a minimal example to realize a NODE).

Recently, Weyl semimetals have emerged as an attractive system to explore various novel electrical and optical phenomena, due to its combination of nontrivial quantum geometry and strong symmetry breaking (to generate Weyl fermions, the system has to break space-inversion, or time-reversal, or both). In the electrical regime, a giant electrical diode effect (the nonreciprocal transport) has been observed in Weyl systems[18]. In the optical regime, novel optical diode effect has

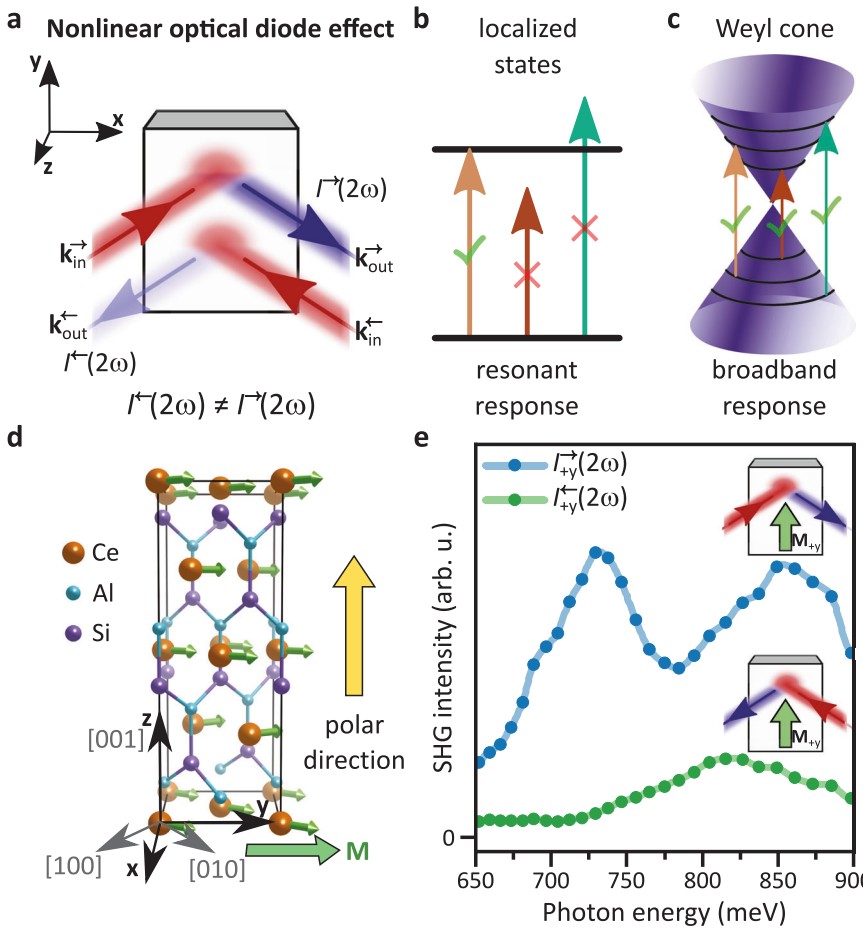

**Fig. 1 | Observation of a broadband NODE in the magnetic Weyl semimetal CeAlSi. a** Definition of the nonlinear optical diode effect (NODE) in reflection. Reversing the light path by swapping light source and detector corresponds to changing the light $\mathbf{k}$ vectors $\mathbf{k}_{in}^{\rightarrow}$ and $\mathbf{k}_{out}^{\rightarrow}$ to $\mathbf{k}_{in}^{\leftarrow}$ and $\mathbf{k}_{out}^{\leftarrow}$. A NODE is evidenced by a difference between intensities $I^{\rightarrow}(2\omega)$ in forward direction and $I^{\leftarrow}(2\omega)$ in reversed direction. **b** For a transition between localized electronic states, such as in wide-gap insulators, only specific photon energies can drive the transition resulting in a

resonant response. **c** For dispersive bands such as Weyl cones, different photon energies can drive transitions leading to a broadband response. **d** Crystal structure of the magnetic Weyl semimetal CeAlSi. We define $+\hat{x} = [110]$, $+\hat{y} = [\bar{1}10]$, and $+\hat{z} = [001]$ relative to the tetragonal unit cell of (paramagnetic) CeAlSi. **e** Observation of a broadband NODE in the $\mathbf{M}_{+y}$ state of CeAlSi. $I_{+y}^{\rightarrow}(2\omega) \gg I_{+y}^{\leftarrow}(2\omega)$ over a broad range > 250 meV.

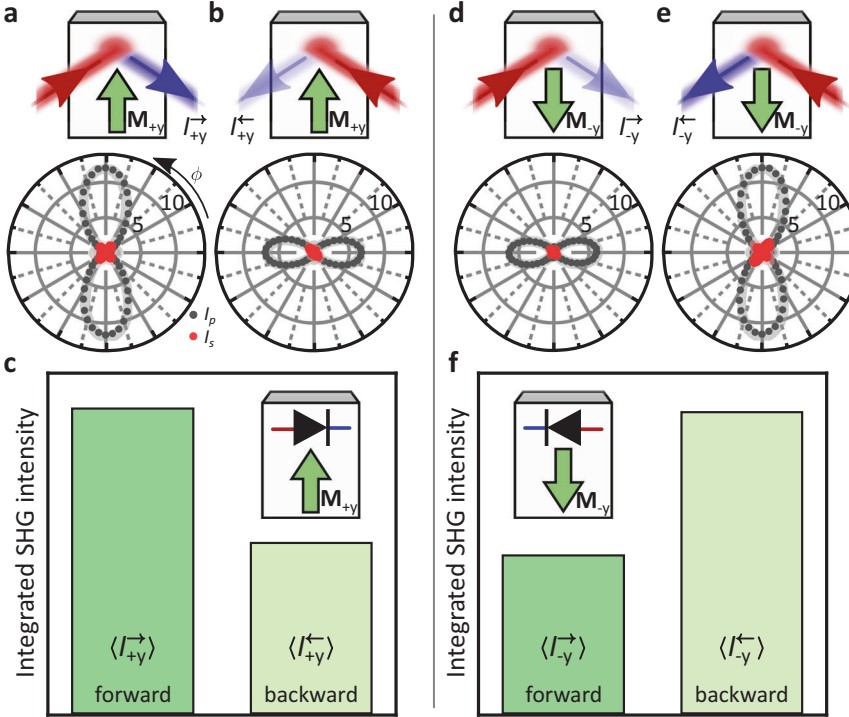

**Fig. 2 | Magnetic control of the NODE. a, b** Polarization-dependent SHG for both propagation directions of the $\mathbf{M}_{+y}$ state (fundamental photon energy 715 meV). **c** Integrated SHG intensity of the $\mathbf{M}_{+y}$ state for both propagation directions. The integrated SHG intensity is defined as the intensity measured without polarization analysis, thus simultaneously detecting s- and p-polarized SHG light, and averaging over all polarizations of the incident fundamental light (see Supplementary Section 8 for details). **d–f** Analogous to panels a-c but for the $\mathbf{M}_{-y}$ state. Solid lines are fits with one consistent set of ED-SHG $\mathcal{X}$ tensor components (see Supplementary Sections 1D and 2 for details). The magnetic state of CeAlSi controls the directionality of the NODE enabling a switchable nonlinear optical diode.

been theoretically considered[1,2,19] but never probed experimentally. In contrast to conventional magnetoelectric insulators[13,14] (Fig. 1a), the Weyl semimetals host linearly dispersive Weyl cones, which can allow for a wide range of photon energies to excite interband optical transitions in the material (Fig. 1b).

In this paper, we report the observation and manipulation of a giant, broadband NODE in the magnetic Weyl semimetal CeAlSi. We demonstrate that at least a six-fold change in SHG intensity is achievable between forward and backward propagating light over a broad spectral range covering 250 meV. Supported by first-principles calculations, we establish a link between the broadband characteristics and the linearly dispersive Weyl fermions in CeAlSi. We show that the directionality of the NODE is directly related to the magnetic order in CeAlSi. We can switch the magnetic order by passing a current and thus demonstrate electrical control of the NODE. We reveal that this basic opto-spintronic functionality persists in technologically relevant, micromachined device structures with the additional advantage of drastically reduced switching currents.

## Results

### Observation of the nonlinear optical diode effect in CeAlSi

CeAlSi belongs to the family of Weyl semimetals with chemical formula $R\text{Al}X$ ($R$ = La-Nd and $X$ = Si or Ge) that recently gained considerable attention as magnetic Weyl semimetals[20–31]. This family is closely related to TaAs[32,33], as they have the same noncentrosymmetric, tetragonal crystal structure (point group $4mm$) and the same valence condition ($R^{3+}[\text{Al}X]^{3-}$ vs. $\text{Ta}^{3+}\text{As}^{3-}$). The introduction of magnetic rare earth elements (Ce, Pr, or Nd) in $R\text{Al}X$ compounds leads to long-range magnetic order. Depending on the rare earth element, a variety of magnetic structures ranging from simple collinear ferromagnetism to noncollinear magnetism and complex multi-$\mathbf{k}$ helical spin structures has been found[21–27]. The interplay between Weyl fermions, inversion

symmetry breaking, and magnetism provides a fertile ground to discover novel emergent quantum electromagnetism in this class of Weyl semimetals.

CeAlSi exhibits a canted ferromagnetic order below $T_C \sim 8.4$ K (Fig. 1c). The net magnetization points along one of the four symmetry-equivalent in-plane crystallographic directions, $\pm$ [110] and $\pm[\bar{1}10]$ (We define $[110] = +\hat{\mathbf{x}}$, $[\bar{1}10] = +\hat{\mathbf{y}}$, and $[001] = +\hat{\mathbf{z}}$, Fig. 1c). Hence, there are four distinct ferromagnetic states, denoted as $\mathbf{M}_{+x}$, $\mathbf{M}_{-x}$, $\mathbf{M}_{+y}$, and $\mathbf{M}_{-y}$. Such a $\mathcal{T}$-breaking magnetic ordering on a $\mathcal{P}$-breaking crystal structure makes CeAlSi a promising candidate for the observation of the NODE.

We consider the NODE explicitly by experimentally reversing the light path. As CeAlSi is semimetallic and therefore highly reflective, we consider the NODE here in a reflection geometry (Fig. 1d). We study in particular SHG from the (001) facet of a CeAlSi single crystal. Therefore, all four magnetic states of CeAlSi exhibit an in-plane net magnetization. We orient the crystal such that the s-polarization of light is parallel to the net magnetization of two out of the four possible magnetic states. We denote those states as $\mathbf{M}_{\pm y}$ states and the states with net magnetization perpendicular to the s-polarization $\mathbf{M}_{\pm x}$ states (see coordinate system in Fig. 1d).

In Fig. 1e, we directly compare the spectra for the recorded SHG intensity for the forward propagation direction ($I^{\rightarrow}(2\omega)$) and the backward propagation direction ($I^{\leftarrow}(2\omega)$) for the magnetic $\mathbf{M}_{+y}$ state. The SHG intensity $I^{\rightarrow}(2\omega)$ in forward direction is more than twice as large as the intensity $I^{\leftarrow}(2\omega)$ in backward direction over the entire considered spectral range of the incident light from 650 meV to 900 meV. Reversing the magnetization reverses the situation (Supplementary Section 4A). We thus find that the SHG response exhibits a NODE over a wide range of fundamental photon energies spanning at least 250 meV.

In Fig. 2a, b we show the polarization dependence of $I^{\rightarrow}_{+y}$ and $I^{\leftarrow}_{+y}$, respectively, at a fundamental photon energy of 715 meV for the magnetic $\mathbf{M}_{+y}$ state of CeAlSi. The polarization dependence also clearly

reveals the NODE in the SHG response. To further corroborate the nonreciprocal character of the observed SHG, we define the integrated SHG intensity $\langle I \rangle$ as the intensity measured without polarization analysis, thus simultaneously detecting s- and p-polarized SHG light, and averaging over all polarizations of the incident fundamental light. Comparing the integrated SHG intensity between the forward and backward direction gives us a measure of the diode effect independent of the light polarization (i.e., the diode effect for unpolarized light). Interestingly, we find for the magnetic $M_{+y}$ state that the integrated SHG intensity $\langle I_{+y}^{\rightarrow} \rangle$ in the forward direction is significantly higher than $\langle I_{+y}^{\leftarrow} \rangle$ in the reversed direction (Fig. 2c). This shows that the NODE is not restricted to particular polarization configurations but a net diode effect can persist even in the case of unpolarized incident light.

## Magnetic control of the NODE

Interestingly, the directionality of the NODE can be controlled by the magnetic order. In Fig. 2a–e, we compare the results for the $M_{+y}$ and $M_{-y}$ states, from which we see that the directionality of the NODE is flipped as we flip the magnetization. Thus, while the broken $\mathcal{P}$ symmetry in general activates the NODE, the broken $\mathcal{T}$ symmetry enables controlling its directionality. By selecting either the $M_{+y}$ or $M_{-y}$ state, we can deterministically set the propagation direction that generates higher integrated SHG intensity (see also Supplementary Movies 1 and 2 for the domain evolution during magnetic-field induced switching).

We can understand the NODE and its magnetic control by the interference of independent SHG tensor components with different origins. In our case, we can use $\chi_{xxx}$ and $\chi_{xxz}$. $\chi_{xxz}$ arises from the noncentrosymmetric polar lattice of CeAlSi; $\chi_{xxx}$ arises from the magnetic order. As shown in Supplementary Fig. 5, the total SHG intensity is given by $|\chi_{xxx}E_x^2 + \chi_{xxz}E_xE_z|^2$. Considering p-polarized incident light, the electric field $\mathbf{E}$ of the light wave is parallel to $\mathbf{k} \times \hat{\mathbf{y}}$ leading to a $\mathbf{k}$ dependence of the SHG response. Specifically, as we reverse the light propagation direction, we reverse the relative sign between $E_x$ and $E_z$. Hence, we find $I_{+y}^{\rightarrow}(2\omega) \propto |\chi_{xxx} + \chi_{xxz}|^2$ and $I_{+y}^{\leftarrow}(2\omega) \propto |\chi_{xxx} - \chi_{xxz}|^2$. Thus, the NODE is based on a directionally dependent mixing between tensor components. It is interesting to note that the origin of the symmetry breaking that enables those tensor components is not crucial. This is in contrast to the established SHG interference imaging technique which strictly relies on interference between order-parameter dependent and order-parameter independent $\chi$ tensor components[17]. As a consequence, a NODE can in principle exist even in non-magnetic materials. Here, however, we explicitly utilize the different origin of $\chi_{xxx}$ and $\chi_{xxz}$ to control the directionality of the NODE. As we reverse the magnetization direction, we reverse the sign of the magnetic SHG tensor $\chi_{xxx}$ but the crystalline SHG tensor component $\chi_{xxz}$ remains unchanged. Thus, we find $I_{-y}^{\rightarrow}(2\omega) \propto |-\chi_{xxx} + \chi_{xxz}|^2$ and $I_{-y}^{\leftarrow}(2\omega) \propto |-\chi_{xxx} - \chi_{xxz}|^2$, which explains why the NODE directionality flips as we flip the magnetization direction.

## Electrical control of the NODE

The noncentrosymmetric symmetry of CeAlSi supports a magneto-electric coupling between the magnetization $\mathbf{M}$ and a current $\mathbf{J}$. Microscopically, such a coupling may be either intrinsic[34,35], mediated by strain[36], or relying on the spin-Hall effect[37] and assisted by the Oersted field of the current[38,39]. In all cases, the magnetoelectric coupling is such that a current in the $xy$-plane favors a magnetization perpendicular to the current in the $xy$-plane with a fixed handedness as depicted in Fig. 3b, providing a way to control the magnetic domain configuration through an electrical current.

We supply a current in the $xy$-plane through a pattern of gold electrodes on a (001) facet of the sample as illustrated in Fig. 3b. We can then use SHG to optically detect the magnetization state. Specifically, we choose s-polarized incident light at 715 meV and detect

p-polarized SHG, where $M_{\pm y}$ states correspond to high and low SHG intensity, respectively (Fig. 2a, b. See also Supplementary Section 5 for more systematic characterizations).

Remarkably, we find that passing a current in CeAlSi can directly choose its magnetization direction (see also Supplementary Movie 3 for *operando* imaging of the domain pattern). Moreover, the selected magnetization persists when the current is removed (i.e., the control is nonvolatile). Specifically, as shown in Fig. 3c, if we pass a negative current of −100 mA along the $\hat{\mathbf{x}}$ direction and then remove this current, we control the magnetization to be along $+\hat{\mathbf{y}}$ ($M_{+y}$); if we pass a positive current of +100 mA and then remove this current, we control the magnetization to be along $-\hat{\mathbf{y}}$ ($M_{-y}$). Therefore, the data in Fig. 3c shows that the magnetization of CeAlSi can be controlled by current. Moreover, Fig. 3d shows the results over the course of >10 consecutive current cycles along opposite directions. We found that the control is highly deterministic, i.e., $-\hat{\mathbf{x}}$ current selects $+\hat{\mathbf{y}}$ magnetization whereas $+\hat{\mathbf{x}}$ current selects $-\hat{\mathbf{y}}$ magnetization. The control is also clearly nonvolatile, i.e., the selected magnetization persists when the current is reduced to zero. By electrically controlling the NODE, we realize here a basic directional nonlinear opto-spintronic functionality.

We note that the above results are obtained on a large mm-sized single crystal. So although the total current needed for the magnetic control appears large (100 mA), the current density is very small. However, as the contacts are deposited on the top surface (Fig. 3b), the spatial distribution of current flow is likely to be inhomogeneous for such a thick bulk crystal (especially along the $z$ direction), so it is hard to reliably estimate the current density. In addition, although the sample remains in the magnetically ordered phase even at the applied highest currents (Supplementary Section 3), we notice significant Joule heating. In order to minimize Joule heating and at the same time reliably estimate the current density, it is highly desirable to fabricate a miniaturized device. Such a micro-device also helps us to explore the tantalizing potential of electrically controlling the nonlinear optical response of CeAlSi for spintronic applications.

In Fig. 3f, we use focused-ion-beam (FIB) milling to fabricate such a micro-device of CeAlSi. The core part of the structure is a free-standing slab of CeAlSi of 126 μm × 29 μm × 2 μm. In order to counteract the significant magnetic shape anisotropy due to the large aspect ratio and keep the device switchable, we prepared the slab with the long axis along the magnetic hard axis, i.e. at an angle of 45° relative to the $x$ and $y$ directions. The device can be poled into a magnetic single-domain state with magnetic fields of about 30 mT. Analogously to the previous bulk measurements, the four magnetic states can be distinguished by their SHG polarization dependence (see also Supplementary Section 9).

Most strikingly, in the absence of external magnetic fields, we can electrically control the magnetic state of the micro-device with dramatically reduced currents. As shown in Fig. 3g, we find that we can change the magnetic state in the FIB device with currents as low as 3 mA corresponding to current densities of around 5 kA/cm² in the slab (in this case, we do not see remanent switching possibly due to shape anisotropy and residual strain in the sample[36,40,41]). The magnetic switching is clearly evidenced by a change of the SHG polarization dependence indicative of switching between the $M_{-x}$ state at −3 mA and a $M_{+x}$ state at +3 mA (Supplementary Section 9). Current-induced Joule heating is negligble at such low currents (Supplementary Section 9). Such a reversible electrical control of a magnetic state may be highly desirable for novel device concepts.

## Observation of a broadband NODE

Note that so far we observed a pronounced NODE for a specific SHG component at 715 meV without deliberate optimization with respect to polarization or photon energy. However, a comparison of Fig. 2a and b reveals that changing the magnetization induces large changes of SHG

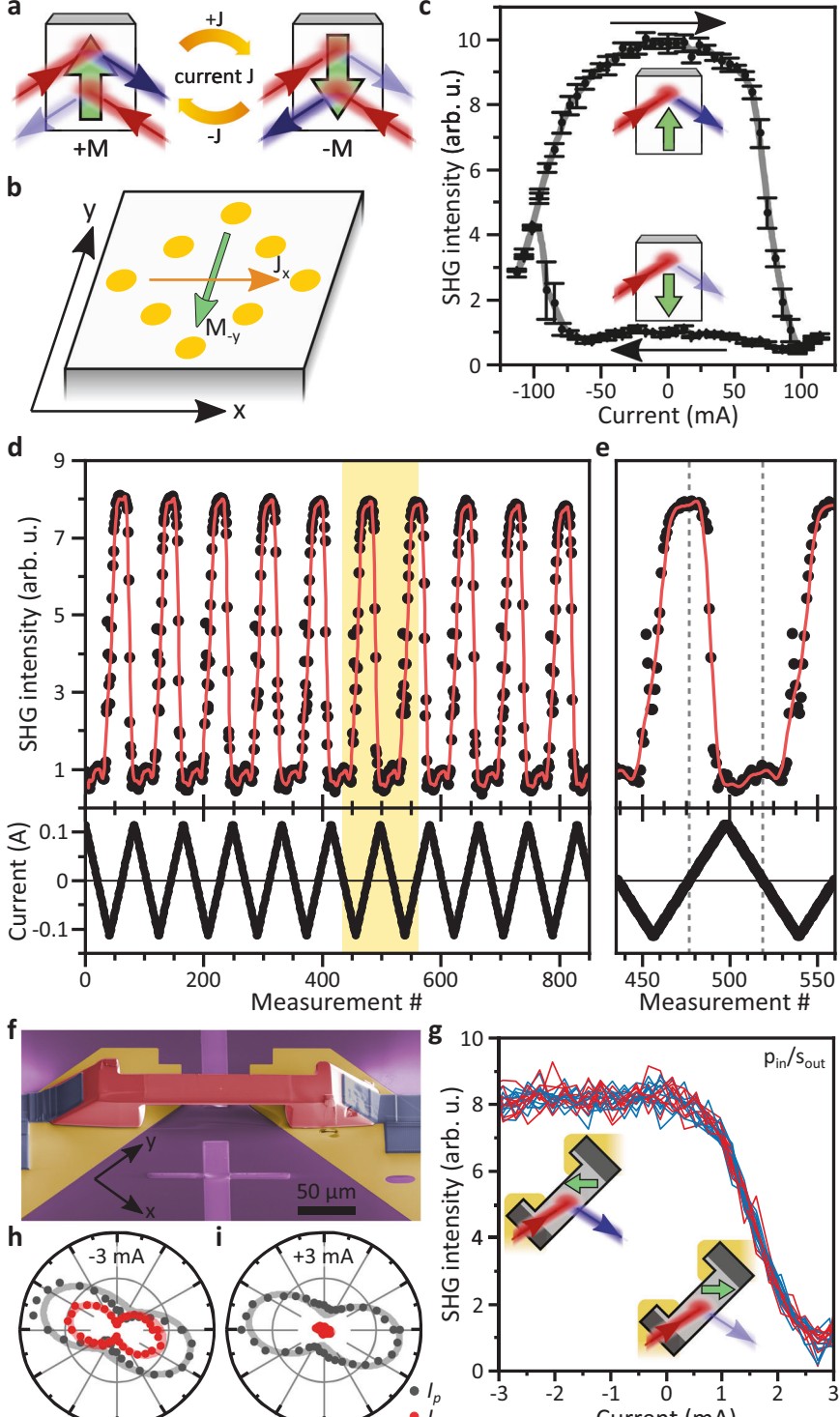

**Fig. 3 | Electrical control of the NODE. a** Schematic of a device concept. Current-induced magnetization switching enables electrical control of the NODE. **b** We deposit eight gold electrodes on the top surface of a bulk mm-sized CeAlSi crystal, which allows us to pass current through the sample. Here, we focus on current along the $\hat{x}$ direction, which causes magnetization switching along the $\hat{y}$ direction. **c** Current hysteresis loop. A current of 100 mA fully switches the magnetization. We measure p-polarized SHG in forward direction with s-polarized incident light. Therefore, high and low intensity correspond to $\mathbf{M}_{+y}$ and $\mathbf{M}_{-y}$ states, respectively. Error bars denote the standard deviation of three consecutive measurements. **d** Current-induced magnetization switching is highly reproducible. Here, we show 10 consecutive cycles. **e** Enlarged view of the highlighted region in d. Dashed lines

correspond to the remanent states. **f** Scanning electron microscopy image of a micro-sized device manufactured by focused-ion-beam milling of CeAlSi (red) on gold electrodes (gold). Platinum contacts (gray) ensure good electrical connections. **g** Currents as small as 3 mA can control the magnetization state. Red and blue curves correspond to consecutive measurements with increasing and decreasing current, respectively. We detect s-polarized SHG, which can discriminate $\mathbf{M}_{\pm x}$ states. A hysteresis is suppressed possibly due to shape anisotropy or residual strain in the sample[36,40,41]. **h, i** SHG polarization dependence under application of -3 mA and +3 mA, respectively. Changes are most prominent for s-polarized SHG (red). Solid lines are fits corresponding to magnetic $\mathbf{M}_{-x}$ and $\mathbf{M}_{+x}$ states, respectively (Supplementary Section 9).

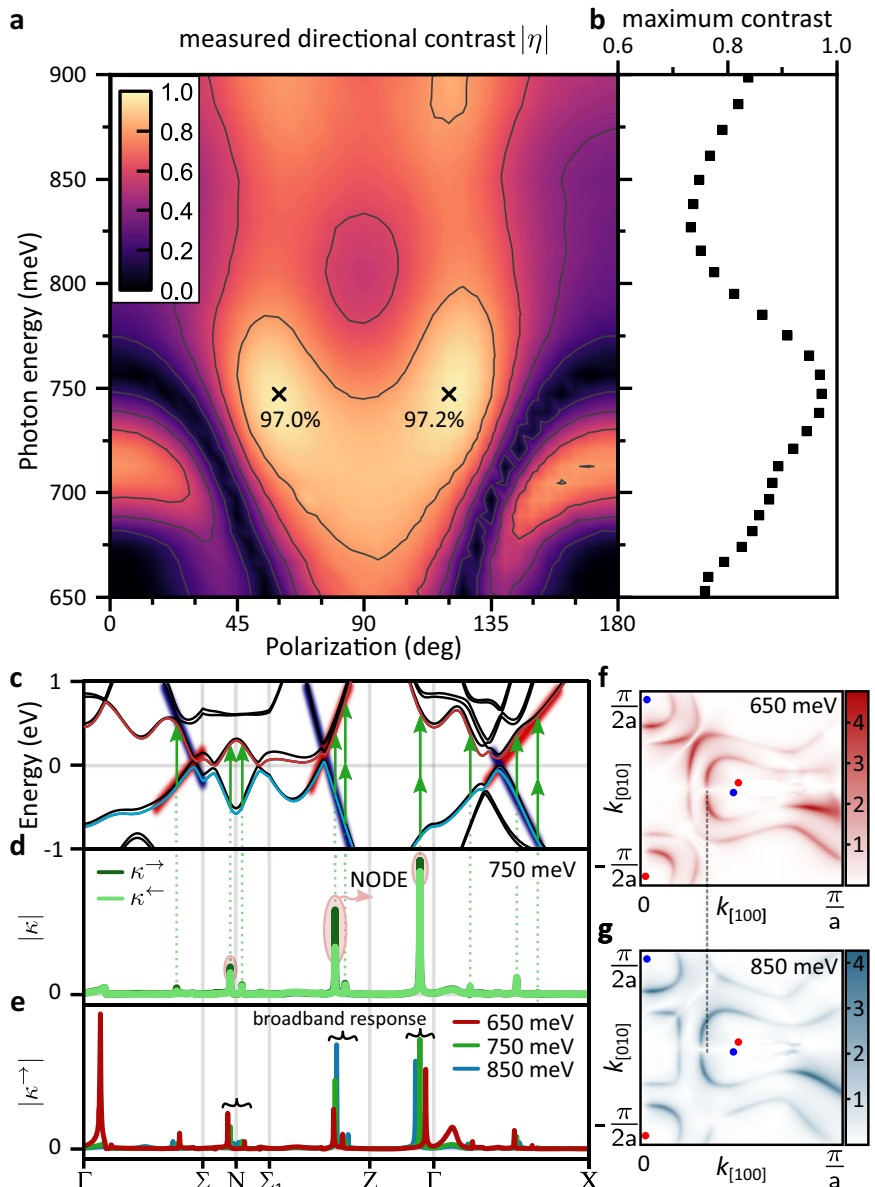

**Fig. 4 | NODE spectroscopy and its microscopic origin from the linearly dispersive bands. a** Measured absolute value of the directional contrast $|\eta|$ between $\mathbf{M}_{\pm y}$ states for p-polarized SHG in forward direction as a function of incident photon energy and polarization. **b** Maximum achievable directional contrast by varying incident polarization as function of photon energy. A contrast of at least 73% corresponding to a six-fold change in SHG intensity is achievable over the whole considered spectral range. **c** Band structure of CeAlSi calculated from first principles in the $\mathbf{M}_{+y}$ state. **d** Momentum-space resolved contributions $\kappa^{\rightarrow}$ and $\kappa^{\leftarrow}$ at 750 meV for forward and backward propagating beams. The difference between $\kappa^{\rightarrow}$ and $\kappa^{\leftarrow}$ leads to the NODE (see text). For clarity, we only show here contributions arising from the valence band marked in blue and the conduction band marked in red. **e** Momentum-space resolved contributions to the SHG intensity in forward direction at incident photon energies $\hbar\omega$ of 650 meV, 750 meV, and 850 meV. Most contributions arise at **k** points where electronic bands are linearly dispersive thus giving rise to a broadband response. **f, g** Distribution of $|\xi_{xxy}|$ (in arbitrary units) in a section of the Brillouin zone plane $k_z = 0.295$ Å$^{-1}$ for $\hbar\omega = 650$ meV and 850 meV, respectively. Red and blue dots indicate the projection of the $W_2$ Weyl nodes of different chirality onto the considered plane. As the photon energy increases, contributions to $|\xi_{xxy}|$ occur at larger distance from the Weyl nodes -- a direct consequence of linearly dispersive bands. Dashed line is a guide to the eye highlighting the expansion of the contribution line with increased photon energy.

intensity for certain polarization components and vanishing changes for others. To quantify the NODE, we define the directional contrast $\eta$ as

$$\eta = \frac{I^{\rightarrow}_{+y} - I^{\leftarrow}_{+y}}{I^{\rightarrow}_{+y} + I^{\leftarrow}_{+y}}. \tag{1}$$

The directional contrast $\eta = 0$ corresponds to absence of the NODE (i.e., the SHG intensity along forward and backward directions is

equal); the directional contrast $\eta = \pm 1$ corresponds to extreme NODE (i.e., SHG can only occur along one direction).

In Fig. 4a, we show the evolution of $|\eta|$ for p-polarized SHG as a function of both incoming laser polarization and SHG photon energy. Here, we compare $I^{\rightarrow}_{+y}$ and $I^{\rightarrow}_{-y}$, which is equivalent to comparing $I^{\rightarrow}_{+y}$ and $I^{\leftarrow}_{+y}$ but experimentally more straightforwardly accessible (see also Supplementary Section 4.C for other assessments of $\eta$). The figure is characterized by a rich behavior that is mirror symmetric along the 0 and 90 deg directions due to mirror symmetries in CeAlSi. At 715 meV we find that $\eta$ exhibits maxima at 0 deg/180 deg (p-polarized light) as

well as near 90 deg (s-polarized light) but vanishes near 45/135 deg, in agreement with Fig. 2a, b. In Fig. 4b, we show the maximum achievable contrast for each SHG photon energy. The directional contrast peaks around 750 meV at 97.2% corresponding to a more than 70-fold change in SHG intensity. Moreover, we find that a directional contrast of at least 73% (corresponding to a more than 6-fold change) is achievable for all photon energies in the considered spectral range between 650 meV and 900 meV.

## Broadband NODE from first principles

In order to understand the microscopic origin of the NODE, we directly compute the NODE from the electronic structure of CeAlSi. Specifically, the NODE is the difference in SHG intensity between opposite propagation directions, which is determined by the two momentum-space resolved quantities $\kappa^{\rightarrow}$ and $\kappa^{\leftarrow}$ as (see methods, Eq. (8)):

$$\text{NODE} = I^{\rightarrow} - I^{\leftarrow} = \left| \int_{BZ} \kappa^{\rightarrow} d^3 \mathbf{k} \right|^2 - \left| \int_{BZ} \kappa^{\leftarrow} d^3 \mathbf{k} \right|^2. \quad (2)$$

Here, $\kappa^{\rightarrow(\leftarrow)} = \sum_{i,j,k} \xi_{ijk} E_j^{\rightarrow(\leftarrow)}(\omega) E_k^{\rightarrow(\leftarrow)}(\omega) A_i^{\rightarrow(\leftarrow)}$, where $\mathbf{E}^{\rightarrow(\leftarrow)}$ is the electric field of the incident light for the forward and backward propagation direction, $\mathbf{A}^{\rightarrow(\leftarrow)}$ is a projector to select either the $s$ or $p$-polarized SHG response, and $\xi_{ijk}$ is the $\mathbf{k}$-space resolved contribution to the nonlinear optical susceptibility ($\xi_{ijk}$ is directly computed from the band structure of CeAlSi, see methods Eqs. (5) and (6)).

Figure 4c shows a section of the band structure of CeAlSi consistent with previous first-principles calculations[24,42] (see methods for further details). Along with the band structure, we show in Fig. 4d the magnitude of $\kappa^{\rightarrow}$ and $\kappa^{\leftarrow}$ in the band structure of CeAlSi (for $\hbar\omega = 750$ meV). We see from Fig. 4d that discrete $\mathbf{k}$ points contribute significantly. By comparing Fig. 4c and d, it is clear that these are the states which allow for optical transitions at $\omega$ or $2\omega$ across the Fermi energy. Importantly, the magnitude of $\kappa^{\rightarrow}$ and $\kappa^{\leftarrow}$ differs significantly, giving rise to the observed NODE.

To understand the band structure origin of the broadband nature of the NODE in CeAlSi, we vary the incident photon energy from 650 meV to 850 meV. We can observe two qualitatively different behaviors (Fig. 4e). On the one hand, we notice a few contributions such as near the $\Gamma$ point along the $\Gamma - \Sigma$ line that occur only at specific photon energies (here 650 meV). The bands at that $\mathbf{k}$ point are relatively flat. As the photon energy changes, such contributions disappear, which is indicative of a typical electronic resonance. Despite the apparent size of this resonant contribution, the spectrum of all SHG tensor components evolves smoothly towards 650 meV indicating an over all negligible effect (Supplementary Section 10). On the other hand, the majority of the SHG response arises at $\mathbf{k}$ points where the electronic bands are linearly dispersive. Accordingly, as we vary the incident photon energy, those contributions only shift slightly to different $\mathbf{k}$ points but remain comparable in magnitude. This observation of SHG originating from linearly dispersive bands is qualitatively different from the conventional resonance picture.

We further elucidate the $k$-space origin of the broadband SHG response in Fig. 4f, g. We consider as an example the distribution of $|\xi_{xxy}|$ in the plane $k_z = 0.295$ Å$^{-1}$. In the paramagnetic phase of CeAlSi, this plane contains the $W_2$ Weyl nodes. In the ferromagnetic phase, the Weyl nodes shift out of the plane. Red and blue dots therefore indicate the projection of the Weyl nodes onto the plane $k_z = 0.295$ Å$^{-1}$. The pair of Weyl nodes in Fig. 4f are enveloped by two lines of strong contributions to $|\xi_{xxy}|$. In analogy to Fig. 4d, we identify the inner and outer lines as transitions that are resonant at $\hbar\omega$ and $2\hbar\omega$, respectively. As we increase the incident photon energy from 650 meV (Fig. 4f) to 850 meV (Fig. 4g), we observe that the lines of strong contributions occur at an increased distance from the Weyl nodes (as indicated by the dashed line). This observation is a direct consequence of the

linearly dispersive bands that disperse over hundreds of meV in CeAlSi. We thus see that the broadband NODE in CeAlSi ultimately arises from the linearly dispersive bands that naturally occur in the vicinity of the Weyl nodes (see also Supplementary Section 10 for an extended discussion).

## Discussion

In summary, we explored the nonlinear optical response of the magnetic Weyl semimetal CeAlSi using SHG spectroscopy. In addition to the crystallographic (time-reversal symmetric) SHG, we observe magnetic (time-reversal asymmetric) SHG, which in a topologically nontrivial material. By physically reversing the light path, we revealed a pronounced NODE.

We introduce the NODE as a novel concept that utilizes a directionally dependent SHG intensity. The effect is rooted in a directionally dependent interference that is conceptually different from the well-known magnetic SHG contrast[17,43,44] or nonreciprocal SHG[13,14]. In particular, the NODE does not strictly require a broken time-reversal symmetry $\mathcal{T}$ nor the simultaneous presence of electric-dipole SHG and magnetic-dipole/electric-quadrupole SHG (which requires a broken spatial inversion symmetry $\mathcal{P}$). Instead, the NODE is microscopically enabled by a broken mirror symmetry (Supplementary Section 1D).

Remarkably, we found a sizable directional contrast of at least 73% over a spectral range exceeding 250 meV − two orders of magnitude wider than previous reports based on nonreciprocal SHG. We note that the directional contrast can in principle be further optimized by varying external parameters like the sample temperature or the angle of incidence.

Microscopically, our DFT calculations show that the broadband NODE is directly related to the electronic band structure. The SHG response emerges from linearly dispersive bands that naturally occur near the Fermi energy in a Weyl semimetal. The linear dispersion allows for strong optical responses over a wide range of frequencies.

Our observations open up a number of intriguing possibilities. First, it suggests the NODE as a powerful method to measure the phase of optical properties. All optical processes (e.g., SHG, Raman) are governed by the corresponding susceptibility tensor, which is determined by the material. Hence, inversely, by measuring the SHG, Raman, or other optical process, we can learn about the symmetry and electronic properties of a material. However, typically we only access the magnitude of the susceptibility, whereas the sign or phase is hard to probe. Here, as we showed above, the NODE arises from the directionally dependent mixing between tensor components which provides insights into their relative phases. It would be interesting to test the optical diode effect for other optical processes such as high-harmonic generation[45], Raman scattering[46], and optically generated spin currents[47], where interesting phenomena hinting nonreciprocity have been observed. Second, in addition to searching for new diode effects based on other nonlinear optical processes, our results open up for a search for other materials exhibiting diode effects. So far, all observation of directional SHG responses were demonstrated in noncentrosymmetric, time-reversal-broken materials, although this should not strictly be necessary. It will be interesting to demonstrate the NODE in a time-reversal symmetric (e.g., antiferromagnetic) or inversion-symmetric systems. Third, we note the low-current density associated with the switching of magnetic order in CeAlSi. It would be of interest to perform future studies to understand its microscopic mechanism and to test it on other magnetic and noncentrosymmetric Weyl fermion systems. The low-current switching in combination with the broadband character of the NODE, may enable fundamentally new device concepts for photonic circuits. From a fundamental point of view, our observation of a broadband NODE is testament to the exotic electromagnetic responses that can be discovered in novel quantum magnets.

## Methods

### Crystal growth

Single crystals of CeAlSi were grown by a self-flux method. The starting materials were Ce and Al ingots, and Si pieces, mixed in ratio Ce:Al:Si = 1:10:1 in an alumina crucible. The crucible was sealed in an evacuated quartz tube, and went through the following heating sequence: the sample was heated from 25 °C to 1000 °C at 3 °C min$^{-1}$, stayed for 12 h, cooled to 700 °C at 0.1 °C min$^{-1}$, stayed for 12 h, and finally centrifuged to remove the residual Al flux at 700 °C.

### SHG measurements

All measurements were performed at 3 K unless explicitly stated differently. In particular, all measurements shown in the main text were performed at 3 K.

Second-harmonic generation (SHG) is a nonlinear optical process that describes the interaction of two photons at frequency $\omega$ in a material leading to the re-emission of one photon at frequency $2\omega$. In the lowest-order electric-dipole approximation, the process can be formally expressed as

$$P_i(2\omega) = \sum_{j,k} \mathcal{X}_{ijk} E_j(\omega) E_k(\omega), \tag{3}$$

where $E_j(\omega)$ and $E_k(\omega)$ denote the electric-field components of the incident light wave and $P_i$ the components of the induced nonlinear polarization oscillating at $2\omega$ (indices $i,j,k$ can be $x,y$, or $z$). The process is mediated by the second-order susceptibility tensor $\mathcal{X}_{ijk}$, which can only have non-vanishing components when $\mathcal{P}$ is broken[34].

All SHG measurements were obtained using a laser system by LightConversion consisting of an amplified femtosecond laser (Pharos, photon energy 1.2 eV, maximum power 10 W, maximum repetition rate 100 kHz) in combination with an optical parametric amplifier (Orpheus One HE). In order to avoid heating effects, we reduce the repetition rate of the laser and employ attenuating filters to achieve an average power on the sample of less than 2 mW. For domain imaging (e.g., Fig. 2i), we chose a spot size of approximately 800 μm. For all other measurements, we focused the laser to a spot size of approximately 150 μm. All SHG measurements were performed in reflection under an angle of incidence of 45°. We adjust the laser polarization of the incident fundamental beam using an achromatic half-waveplate, while selectively detecting s- and p-polarized SHG intensity (red and gray data points, respectively) using a Glan-laser polarizer and a thermo-electrically cooled electron-multiplying CCD camera (EMCCD). For spectrally resolved SHG measurements, a grating spectrometer was mounted in front of the EMCCD camera. For all other measurements, the EMCCD camera was mounted behind a narrow band pass filter centered at 1.425 eV (bandwidth 30 meV). We used additional filters directly before the sample to block parasitic SHG signals from all previous optical components as well as directly after the sample to block the fundamental beam.

The measured SHG intensity is proportional to

$$I(2\omega) = \left| \mathbf{P}(2\omega) \cdot \mathbf{A} \right|^2 = \left| \sum_{i,j,k} \mathcal{X}_{ijk} E_j(\omega) E_k(\omega) A_i \right|^2, \tag{4}$$

where $\mathbf{A}$ is the direction of the transmitted polarization axis of the analyzer (here $s$ or $p$ polarization).

### Focused-ion-beam microstructuring

An oriented large single crystal of CeAlSi was milled using a Xe-Plasma FIB (Thermofisher Helios PFIB). At 2.5 mA, 30 kV, first a rectangular bar was milled from the parent crystal (bar length 300 mm, width 50 mm, height 30 mm). This bar was attached with Pt deposition to an in-situ micromanipulator and rotated by 90°, to access the side face of the bar. At this angle, the trapezoidal outer shape and the central bridge were patterned at 500 nA, 30 kV for coarse and 60 nA, 30 kV for fine patterning. Furthermore, the outer feet were polished flat at 60 nA, 30 kV to ensure a flat mating with the substrate. A sapphire substrate (2 × 2 mm) with two large, lithographically prepared Au leads (10 nm Ti/100 nm Au) was introduced into the chamber and the bridge rotated back. Using the in-situ manipulator, the bridge was placed ontop of these Au pads and connected by Pt deposition (60 nA, 12 kV), on the right foot in main Fig. 3f. The tip of the manipulator was cut off (60 nA, 30 kV), and the remains of the manipulator are well visible on the front section of the right foot. Despite best efforts of alignment, an approximately 500 nm gap appeared between the left foot and the Au pad. The micromanipulator was used to gently push the structure flat, yet without forming a solid bond via deposition. Using the same settings as on the right side, the left side was connected with Pt. The sizable depositions resulted in visible overspray, which at these channel lengths does not notably conduct. Out of caution, the overspray was removed in a rectangular channel on both sides of the bridge (60 nA, 30 kV). The FIB process at 30 kV usually results in a 20 nm thick amorphous layer which may cause issues with the SHG experiments. This layer was strongly reduced using low-voltage polishing. At an angle of 52° between the beam and the surface normal, the central top of the bridge was irradiated at 60 nA, 5 kV in a final cleaning step. As the entire process is performed using Xe beams, no implantation of the primary ions is expected.

### First-principles calculations

We performed density-functional theory calculations[48] using the *Vienna Ab initio Simulation Package* (VASP)[49] with the Perdew-Berke-Ernzerhof's (PBE) pseudopotential[50] in the generalized-gradient approximation[51]. We adopted $U_{\text{eff}} = 6$ eV for the Hubbard U-term acting on the Ce $f$-orbital electrons[24]. The Wannier tight-binding Hamiltonians were established using the *Wannier*90 package[52]. The Wannier bands were symmetrized to restore the symmetry restrictions of the CeAlSi space group[53]. The magnetic order of CeAlSi is described by two magnetic sublattices of equal magnitude with noncollinear magnetic moments $\mathbf{m}_1$ and $\mathbf{m}_2$ on Ce $4f$ orbitals such that $(\mathbf{m}_1 + \mathbf{m}_2) \parallel [110]$. The angle between $\mathbf{m}_1$ and $\mathbf{m}_2$ is 70° as indicated in Ref. 24.

### Calculation of $\chi_{ijk}$

We calculated the nonlinear optical susceptibility $\chi_{ijk}$ for electric-dipole SHG according to the diagrammatic approach to the nonlinear optical response[54,55]. Interband transitions cause two contributions $\xi^I$ and $\xi^{II}$ such that $\chi_{ijk} = \int_{BZ} d^3\mathbf{k} \left( \xi^I_{ijk} + \xi^{II}_{ijk} \right)$, where

$$\xi^I_{ijk} = C \sum_{m \neq n} f_{mn} \left( \frac{h^{ij}_{nm} h^k_{mn} + h^{ik}_{nm} h^j_{mn}}{\omega + i\eta - \omega_{mn}} + \frac{h^{jk}_{mn} h^i_{nm}}{2\omega + i\eta - \omega_{mn}} \right), \tag{5}$$

$$\xi^{II}_{ijk} = C \sum_{m \neq n \neq p} \frac{h^i_{pm} \left( h^j_{mn} h^k_{np} + h^k_{mn} h^j_{np} \right)}{\omega_{mn} + \omega_{np}}$$
$$\left( \frac{f_{np}}{\omega + i\eta - \omega_{pn}} + \frac{f_{nm}}{\omega + i\eta - \omega_{nm}} + \frac{2f_{mp}}{\omega + i\eta - \omega_{pm}} \right) \tag{6}$$

Here, $\{i,j,k\}$ run over $\{x,y,z\}$ directions, $C = \frac{e^3}{2i\epsilon_0 \hbar^2 \omega^3}$, $\{m,n,p\}$ are the band indices, $h^i_{mn} = \frac{1}{\hbar}\langle m | \partial_{k_i} h(\mathbf{k}) | n \rangle$, $h^{ij}_{mn} = \frac{1}{\hbar}\langle m | \partial_{k_i} \partial_{k_j} h(\mathbf{k}) | n \rangle$, $\omega_{mn} = \frac{1}{\hbar}(\epsilon_m - \epsilon_n)$ represents the transition frequency between bands $m$ and $n$, $f_{mn}$ is the difference in band occupation according to the Fermi-Dirac distribution, and $\epsilon_0$ is the vacuum permittivity. The derivatives $h^i_{mn}$ and $h^{ij}_{mn}$ can be rewritten in terms of the generalized Berry connection

$\mathcal{A}_{mn}^i = i\langle m|\partial_{k_i}|n\rangle$[56]:

$$h_{mn}^i = \frac{1}{\hbar}\langle m|\partial_{k_i}h(\mathbf{k})|n\rangle = \partial_{k_i}\langle m|h(\mathbf{k})|n\rangle - \langle\partial_{k_i}m|h(\mathbf{k})|n\rangle - \langle m|h(\mathbf{k})|\partial_{k_i}n\rangle$$
$$= \delta_{mn}\partial_{k_i}\epsilon_m + i\hbar\omega_{mn}\mathcal{A}_{mn}^i.$$

$$(7)$$

As the summation in Eqs. (5) and (6) only considers interband transitions ($m \neq n$), we find $h_{mn}^i = i\hbar\omega_{mn}\mathcal{A}_{mn}^i$. Similarly, we find $h_{mn}^{ij} = i\hbar\mathcal{A}_{mn}^i\partial_{k_j}\omega_{mn}$ for the relevant case of $m \neq n$. Therefore, all interband contributions to $\chi_{ijk}$ can be related to the generalized Berry connection $\mathcal{A}_{mn}^i$.

Using $\chi_{ijk} = \int_{BZ} d^3\mathbf{k}\,\xi_{ijk}$ and Eq. (4), we can express the SHG intensity as

$$I(2\omega) = \left|\int_{BZ}\sum_{i,j,k}\xi_{ijk}E_j(\omega)E_k(\omega)A_i d^3\mathbf{k}\right|^2 = \left|\int_{BZ}\kappa\,d^3\mathbf{k}\right|^2,$$

$$(8)$$

where we define $\kappa = \sum_{i,j,k}\xi_{ijk}E_j(\omega)E_k(\omega)A_i$.

## Data availability
Source data are provided in the Source Data file. All data are available upon request from the corresponding authors. Source data are provided with this paper.

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

## Acknowledgements

Work in the SYX group was partly supported through the Center for the Advancement of Topological Semimetals (CATS), an Energy Frontier Research Center (EFRC) funded by the U.S. Department of Energy (DOE) Office of Science (fabrication and measurements), through the Ames National Laboratory under contract DE-AC0207CH11358, and partly by the AFOSR grant FA9550-23-1-0040 (data analysis), and partly by the NSF Career DMR-2143177 (manuscript writing). SYX also acknowledges the Corning Fund for Faculty Development. The work in the QM group was supported through the CATS, an EFRC funded by the DOE Office of Science (manuscript writing), through the Ames National Laboratory under contract DE-AC0207CH11358. QM also acknowledges the support from NSF through a CAREER award DMR-2143426 (material supplies) and the CIFAR Azrieli Global Scholars Program. C.T. acknowledges support from the Swiss National Science Foundation under project no. P2EZP2_191801 and from the Harvard University Climate Change Solutions Fund. J.N. acknowledges support from the Swiss National Science Foundation under project no. P2EZP2_195686. F.H. received funding by the National Natural Science Foundation of China under grant 52103353. P.J.W.M. acknowledges funding by the European Research Council (ERC) under the European Union's Horizon 2020 research and innovation programme (MiTopMat, grant agreement no. 715730). C.G. received funding by the Swiss National Science Foundation (grant no. PP00P2_176789). This material is based upon work supported by the Air Force Office of Scientific Research under award number FA2386-21-1-4059. K.T.L. acknowledges the support of HKRGC through Grants RFS2021-6S03, C6025-19G, AoE/P-701/20, 16310520, 16310219 and 16307622.

## Author contributions

C.T. designed and conducted the SHG experiments with assistance from J.-X.Q., H.-C.L., Y.-F.L., A.G., D.B., T.D. and S.-C.H. C.T. evaluated the data with support from J.N. X.-J.G., C.-P.Z., Y.-M.X. and K.T.L. performed the first-principles calculations. H.-Y.Y., Y.F., F.H. and F.T. synthesized CeAlSi single crystals. C.G. and P.J.W.M. prepared the FIB cut. C.T., Q.M. and S.-Y.X. wrote the manuscript with discussions and contributions from all authors.

## Competing interests

The authors declare no competing interests.
