## [Peer Review File · Nature Communications]

REVIEWER COMMENTS

Reviewer #1 (Remarks to the Author):

In this manuscript, C. Tzschaschel and colleagues reported the nonlinear optical diode effect (NODE) in a magnetic Weyl semimetal CeAlSi. The authors observed six-factor variations of the measured second-harmonic generation (SHG) intensity by reversing the beam path direction. In addition, the NODE can be directly controlled by the direction of the CeAlSi magnetization, enabling electrical control of SHG intensity through current-induced magnetization switching. Authors attribute the mechanism of the NODE effect to directionally dependent mixing between SHG tensor components in CeAlSi. Overall, this is a well-written manuscript with interesting and novel experimental measurement results, which are also in agreement with the density-functional theory calculations. The presented work will be interesting to a broad range of research communities of the condensed-matter physics field, such as topological quantum materials, nonlinear optics, and spintronics. I would recommend publication of this manuscript in Nature Communications after authors address the following points:

1. The demonstrated nonlinear optical diode contrast characterized by the variation of the SHG intensity between the forward and backward light is a factor of six in this paper. How to further improve the diode contrast experimentally? Does NODE depend on the wavelength of the light, orders of optical harmonics, and measurement temperatures?
2. This work utilizes a noncentrosymmetric ferromagnetic Weyl semimetal CeAlSi to demonstrate NODE. It will be helpful if authors could comment on whether this effect can be observed in other topological materials such as centrosymmetric Weyl semimetals or antiferromagnetic Weyl semimetals.
3. As shown in Fig. 3(c) in the manuscript, the SHG intensity can be controlled by electrical currents, and the authors attribute this effect to the current-induced magnetization switching in CeAlSi. Have the authors estimated the potential contribution from Joule heating in this experiment? It can be noticed that the difference of SHG intensity between the up and down magnetized states decreases with the increasing electrical currents. What is the reason for that? What is the magnetic state of CeAlSi at the maximum current (~100 mA)?
4. Authors observed the two-peak feature of SHG intensity spectrum in Fig. 1(d) for the forward propagating light (blue curve), while the spectrum for the backward propagating light (green curve) in Fig. 1(d) and the spectra shown in Extended Data Fig. 1(d) exhibit only one peak. Could authors further explain the reason for this difference and the underlying mechanism behind these SHG spectra?

Reviewer #2 (Remarks to the Author):

This manuscript by C. Tzschaschel et al. reports an observation of “nonlinear optical diode effect (NODE)” of second harmonic generation (SHG) in reflection geometry, with current-induced control of diode direction. The authors claim that the broad-band nature of this NODE is due to the Weyl-type linear

dispersion of the electronic states in a magnetic Weyl semimetal CeAlSi. Although the experimental findings are potentially interesting and appreciated, I have some concerns regarding its novelty, and also found possible problems in the interpretation and discussion given in the manuscript, as detailed below. Therefore, I do not recommend this paper to be published in Nature Communications in the present form.

1. Please strictly define the nonreciprocity and diode effect in the main text, possibly with the microscopic viewpoint. This is very important to avoid future confusions in this field. I believe that these notions are related to the space-inversion and time-reversal operations, whereas in the present case the experiments are designed for the ~ 90 -degrees rotation of the incident light path.

Physically, the observed modulation of the SHG intensity comes from the interference between the lattice-originated SHG and the magnetic-order-induced one (possibly both are electric dipole radiations), which has been reported in several papers already (see, for example, V. V. Pavlov et al., Phys. Solid State 45, 662 (2003) and S. Toyoda et. al., Phys. Rev. Mater. 7, 024403 (2023)). Importantly, these observations are not classified as (and not claimed to be) “optical nonreciprocity”, because electric dipole transition is an even function for the reversal of light momentum k . Previous works (refs. 12, 13) showing nonlinear optical diode effects utilize the interference between electric dipole SHG and magnetic dipole (or possibly quadrupole) one, with the latter changing sign upon the reversal of k .

At this stage, I’m not fully convinced that the observed phenomena can be referred to as “nonreciprocal” effect. If supposing that the sample is enough thin to have a transmission SHG signal, the nonreciprocity still survives?

2. The broadband response is emphasized in the paper and discussed in relation to the linear dispersion of Weyl fermion in CeAlSi. However, generally, the interference of two electric dipole transitions in the SHG process can be broadband, as demonstrated in the references above.

Given that the broadband nature of the physical process discussed has been already reported in conventional oxides, meaning that the linear dispersion of Weyl electrons seems to be not necessary, the novelty of the present manuscript would not be so high.

3. In Fig. 1, please compare the experimental SHG spectra and DFT one directly. This is necessary to confirm the quality of these data.

4. Fig. 2: Please show the incident angle dependence. It should be possible to experimentally deduce the amplitude of each tensor elements, to understand the details of interference.

5. In the movie of current-induced control, the domain structures change largely even outside of the current path. Please elaborate this point. I think, discussions (on the effective magnetic field etc.) should be given in the main text, not in the figure caption.

Minor:

6. Experimental conditions are not fully described. For example, it is not clear that at which temperature

the experiments are executed.

7. Fig. 1 caption: Please define what is “localized electronic transition” means here.

8. As for the experiments on the microdevice (Fig. 3h and Extended data Fig. 7), please explain which tensor elements are active, to analyze the observed anisotropy pattern.

General Remark

We thank the reviewers for the careful reading of our manuscript. We are thrilled to see that both reviewers expressed a positive assessment. Reviewer #1 emphasized that *“this is a well-written manuscript with interesting and novel experimental measurement results, which are also in agreement with the density-functional theory calculations. The presented work will be interesting to a broad range of research communities of the condensed-matter physics field”*. Reviewer #2 *“appreciated”* our results and found them *“potentially interesting”*.

Both reviewers also raise highly insightful and important questions that we are happy to address. In particular, Reviewer #1 suggested to discuss the generalizability of our observations with respect to other external parameters or other material systems. Reviewer #2 asked us to clarify the relationship between our work and previous observations on nonreciprocal SHG and magnetic contrast. We have very carefully addressed all questions. Thanks to these insightful comments from the reviewers, the clarity and quality of the manuscript are greatly improved. Most importantly, we now provide extensive supplementary information containing a wide range of additional systematic measurements and analyses that substantiate our claims.

In particular, we

- Discuss the symmetry of our newly observed nonlinear optical diode effect (NODE). We show that the symmetry requirements for the occurrence of the NODE are different from the requirements for previously observed nonreciprocal SHG and magnetic contrast, which allows us to distinguish the three phenomena on a microscopic level.
- Present systematic measurements as a function of temperature, magnetic field, laser polarization, photon energy, propagation direction, and current density.
- Provide an in-depth understanding of the reported SHG signals including a quantification of the involved SHG tensor components and a discussion of the influence of external parameters such as angle of incidence, sample temperature, and photon energy on the magnitude of the NODE.
- Include k-space resolved density-functional calculations to clarify the connection between our experimental observations and the electronic band structure of CeAlSi.

We hope that the reviewers find our answers thorough, reasonable and clear.

Below, we will provide a detailed point-by-point response to all concerns raised by the reviewers. We display the reviewer comments in blue and our response in black. Changes in the main text are indicated in red.

Point-by-point response

Reviewer #1:

In this manuscript, C. Tzschaschel and colleagues reported the nonlinear optical diode effect (NODE) in a magnetic Weyl semimetal CeAlSi. The authors observed six-factor variations of the measured second-harmonic generation (SHG) intensity by reversing the beam path direction. In addition, the NODE can be directly controlled by the direction of the CeAlSi magnetization, enabling electrical control of SHG intensity through current-induced magnetization switching. Authors attribute the mechanism of the NODE effect to directionally dependent mixing between SHG tensor components in CeAlSi. Overall, this is a well-written manuscript with interesting and novel experimental measurement results, which are also in agreement with the density-functional theory calculations. The presented work will be interesting to a broad range of research communities of the condensed-matter physics field, such as topological quantum materials, nonlinear optics, and spintronics. I would recommend publication of this manuscript in Nature Communications after authors address the following points:

We greatly appreciate the reviewer's succinct summary and positive assessment of our work.

1.1. The demonstrated nonlinear optical diode contrast characterized by the variation of the SHG intensity between the forward and backward light is a factor of six in this paper. How to further improve the diode contrast experimentally? Does NODE depend on the wavelength of the light, orders of optical harmonics, and measurement temperatures?

We fully agree with the reviewer's sentiment. While we introduce the NODE in our work, important questions for follow-up studies can evolve around optimizing the NODE for potential applications. The NODE can be optimized with respect to any parameter that changes the interference between different SHG tensor components.

These include:

- **Wavelength:** The NODE does in fact depend on the wavelength of the light as demonstrated in Fig. 4a. While we observed the NODE specifically for second-harmonic generation, a nonlinear optical diode effect is certainly also possible for other optical harmonics with, in general, different spectral dependencies.
- **Temperature:** The NODE does further depend on temperature, as can be seen from Supplementary Fig. 4: Comparing for instance panels a and f (measured at $3\text{K} < T_C$), the polarization dependencies reveal a clear NODE, whereas no NODE is observable at temperatures above T_C (panels e and j). As all SHG tensor components evolve smoothly as a function of temperature across a magnetic phase transition, the NODE must evolve smoothly, too. Therefore, varying the temperature will change the NODE.
- **Angle of incidence:** The NODE does also depend on the angle of incidence. By symmetry, the NODE cannot be observed in a normal incidence reflection geometry (because reversing the light path does not change the experiment). As the mixing of different SHG tensor components changes smoothly as a function of incidence angle, the NODE must evolve smoothly, too.

In fact, based on the nonlinear optical susceptibilities that we can extract from our polarization-resolved SHG measurements (Supplementary Section 2), we can calculate the NODE for all incident angles. In Figure R1 below, we show the calculated SHG intensity and directional contrast for s-polarized incident light, p-polarized SHG in the magnetic M_{+y} state of CeAlSi. Experiments were

performed at +/- 45 deg incident angle, but calculations indicate that the directional contrast could be improved to >99.9% by adjusting the angle of incidence.

Figure R1 **Incident angle dependence.** Calculated incident angle dependence based on the relative magnitudes and phases of the SHG tensor component extracted from experimental polarization dependencies. An angle of incidence of 45° is not optimal. By adjusting the angle of incidence, a numerically determined directional contrast >99.9% could be possible (inset). Black circles mark the experimental configuration of +/- 45°. This figure is also shown as Supplementary Fig. 7 in the Supplementary Information.

Even without any of such deliberate optimizations, we already observed a significant broadband NODE, where the measured SHG intensity can change by *at least* a factor of six between forward and backward propagation over a wide bandwidth exceeding 250 meV. By optimizing with respect to the wavelength and **polarization**, we demonstrate a more than 70-fold change of SHG intensity (Fig. 4a).

We now emphasize the potential for further optimization in the discussion section of the main text (lines 258 - 261): “**Remarkably, we found a sizable directional contrast of at least 73% over a spectral range exceeding 250 meV --- two orders of magnitude wider than previous reports based on nonreciprocal SHG. We note that the directional contrast can in principle be further optimized by varying external parameters like the sample temperature or the angle of incidence.**”

1.2. This work utilizes a noncentrosymmetric ferromagnetic Weyl semimetal CeAlSi to demonstrate NODE. It will be helpful if authors could comment on whether this effect can be observed in other topological materials such as centrosymmetric Weyl semimetals or antiferromagnetic Weyl semimetals.

We thank the reviewer for raising the important point of generalizability.

As the reviewer correctly pointed out, we utilized the noncentrosymmetric magnetic Weyl semimetal CeAlSi in our work. This material breaks both spatial inversion symmetry \mathcal{P} and time-reversal symmetry \mathcal{T} . Interestingly, neither of those is required for the presence of the NODE.

Qualitatively, the NODE describes a difference in SHG intensity between two opposing light propagation directions. As the two propagation directions can be related by a mirror operation \mathcal{M} , the NODE relies on a breaking of that mirror symmetry \mathcal{M} . We therefore expect the NODE to be an extremely general effect that can occur in ferromagnetic semimetals (like CeAlSi or other RAIX compounds), antiferromagnetic semimetals (like CuMnAs or Mn₂Au) or even nonmagnetic semimetals

(like TaAs). Moreover, we can expect a NODE in both noncentrosymmetric materials (like CeAlSi) as well as centrosymmetric semimetals (like Mn₃Sn).

We would like to note that a noncentrosymmetric, ferromagnetic Weyl semimetal like CeAlSi is a prime candidate material for the first demonstration of the NODE. Firstly, due to its noncentrosymmetric nature, CeAlSi allows for a strong electric-dipole-type SHG response, which is symmetry-forbidden in centrosymmetric materials. Secondly, the weak ferromagnetic order in CeAlSi allows us to demonstrate control of the directionality of the diode effect by means of a magnetic field and currents. This is possible as the relevant mirror symmetry is broken by the magnetic order. Switching the magnetization thus affects only the “mirror odd” tensor components but leaves the “mirror even” components unchanged, which results in a switch of the directionality of the diode effect. In nonmagnetic materials, the NODE will be build-in and the directionality cannot easily be switched. Thirdly, the linearly dispersive bands that naturally occur in the Weyl semimetal CeAlSi give rise to an extreme bandwidth of the NODE.

It is worth noting that magnetic-dipole and electric-quadrupole SHG contributions are allowed even in centrosymmetric materials. While the absolute SHG intensity of those contributions is typically much weaker, they can still give rise to a NODE (i.e., a directional contrast) that may be equally pronounced as a NODE based on electric-dipole SHG in noncentrosymmetric materials.

We now mention the exciting possibility to observe a NODE in systems with higher symmetry in the discussion section of the main text (lines 273 - 278): “in addition to searching for new diode effects based on other nonlinear optical processes, our results open up for a search for other materials exhibiting diode effects. So far, all observation of directional SHG responses were demonstrated in noncentrosymmetric, time-reversal-broken materials, although this should not strictly be necessary. It will be interesting to demonstrate the NODE in a time-reversal symmetric (e.g., antiferromagnetic) or inversion-symmetric systems.”

1.3. As shown in Fig. 3(c) in the manuscript, the SHG intensity can be controlled by electrical currents, and the authors attribute this effect to the current-induced magnetization switching in CeAlSi. Have the authors estimated the potential contribution from Joule heating in this experiment? It can be noticed that the difference of SHG intensity between the up and down magnetized states decreases with the increasing electrical currents. What is the reason for that? What is the magnetic state of CeAlSi at the maximum current (~100 mA)?

We thank the reviewer for giving us the opportunity to elaborate further on the electrical control of the NODE. The reviewer is exactly right: the effect of Joule heating cannot be neglected at high currents and one needs to be mindful about the magnetic state of the material at the highest currents.

First of all, the sample remains in the magnetic state for all applied currents. This becomes apparent from Fig. 3c as the detected SHG intensity is different for +113 mA and -113 mA. Since reversing the magnetization is equivalent to reversing the propagation direction in CeAlSi, we thus still observe a finite NODE as a contrast between the two opposite magnetic states even at the highest currents.

The approximately parabolic decrease of the SHG intensity that can be seen in Fig. 3c between +/- 50 mA is exactly rooted in Joule heating, as the reviewer suggested.

We now added additional systematic measurements in the Supplementary Information. In particular, Section 3 focuses on temperature dependent characterizations and contains the following figure:

Figure R2 **Temperature dependent characterization.** **a**, SHG intensity as a function of temperature and applied electric DC current. For temperatures below 4K, the sample remains in the ferromagnetically ordered phase for all applied currents. **b**, Temperature dependent SHG intensity in the absence of electric current. We find a magnetic transition temperature of $T_c = 8.2$ K. This figure is part of the Supplementary Information as Supplementary Figure 8.

Focusing here on Figure R2a, we see that Joule heating raises the sample temperature by approximately 4K at ± 113 mA. We also see that the sample remains in the ferromagnetically ordered phase at 3K for even the highest applied currents.

In addition to the new data in Supplementary Section 3, we comment now on the effect of Joule heating in the main text (lines 172 - 176): “In addition, although the sample remains in the magnetically ordered phase even at the applied highest currents (Supplementary Section 3), we notice significant Joule heating. In order to minimize Joule heating and at the same time reliably estimate the current density, it is highly desirable to fabricate a miniaturized device.”

1.4. Authors observed the two-peak feature of SHG intensity spectrum in Fig. 1(d) for the forward propagating light (blue curve), while the spectrum for the backward propagating light (green curve) in Fig. 1(d) and the spectra shown in Extended Data Fig. 1(d) exhibit only one peak. Could authors further explain the reason for this difference and the underlying mechanism behind these SHG spectra?

Thank you for this question. Ultimately, the NODE is based on interference between different components of the SHG tensor χ . Such interference can change the magnitude of the SHG intensity, and may also affect the spectral shape. For CeAlSi specifically, there are up to 10 independent, in general complex, SHG tensor components contributing to the SHG intensity. Therefore, it is very difficult for us to pinpoint quantitatively which component is responsible for certain spectral features.

To illustrate qualitatively that interference can change the number of peaks in a spectrum, we can consider the simplified example illustrated in Figure R3 below. We consider the case of two interfering, purely real χ components χ_1 and χ_2 that have the shown spectral dependencies and amplitudes, where χ_1 exhibits one broad Gaussian spectrum and χ_2 exhibits two narrower Gaussian features. Constructive interference between the two contributions yields the spectrum shown in blue below with a two-peak feature. In contrast, destructive interference results in a spectrum that seemingly exhibits only one peak.

Figure R3 **SHG interference model**. Interference between the two χ contributions shown in the upper panel results in the spectra shown in the lower panel. This basic model reproduces many features of the observations in Fig. 1e of the main text.

This simple example reproduces most of the features that we observe in Fig. 1e. By construction, the maximum of the destructive interference spectrum at 800 meV coincides with the minimum of the constructive interference spectrum. Allowing for complex χ components with photon-energy-dependent phases would allow for shifts between them.

In summary, depending on the spectral shape of the individual contributions, their interference can exhibit a different number of maxima. Unfortunately, it is not easily possible to relate the arbitrary SHG components χ_1 and χ_2 to specific components in CeAlSi.

Side note: we assumed for this basic model $\chi_1 = e^{-\left(\frac{E-800 \text{ meV}}{150 \text{ meV}}\right)^2}$ and $\chi_2 = 0.8 * e^{-\left(\frac{E-725 \text{ meV}}{40 \text{ meV}}\right)^2} + 0.8 * e^{-\left(\frac{E-870 \text{ meV}}{40 \text{ meV}}\right)^2} + 0.2$. The lower plot directly shows $(\chi_1 + \chi_2)^2$ and $(\chi_1 - \chi_2)^2$ without any further scaling.

Reviewer #2:

This manuscript by C. Tzschaschel et al. reports an observation of “nonlinear optical diode effect (NODE)” of second harmonic generation (SHG) in reflection geometry, with current-induced control of diode direction. The authors claim that the broad-band nature of this NODE is due to the Weyl-type linear dispersion of the electronic states in a magnetic Weyl semimetal CeAlSi. Although the experimental findings are potentially interesting and appreciated, I have some concerns regarding its novelty, and also found possible problems in the interpretation and discussion given in the manuscript, as detailed below. Therefore, I do not recommend this paper to be published in Nature Communications in the present form.

We sincerely thank the reviewer for taking the time to review our paper and for the highly insightful comments. We found the reviewer’s comments extremely helpful. In particular, the reviewer raised important concerns about the distinction between our work on the “nonlinear optical diode effect” and previous groundbreaking works on “nonreciprocal SHG” (now cited as Refs. 13 and 14 in the main text). Below, we will carefully address the reviewer’s comments point-by-point.

2.1. Please strictly define the nonreciprocity and diode effect in the main text, possibly with the microscopic viewpoint. This is very important to avoid future confusions in this field. I believe that these notions are related to the space-inversion and time-reversal operations, whereas in the present case the experiments are designed for the ~ 90 -degrees rotation of the incident light path.

Physically, the observed modulation of the SHG intensity comes from the interference between the lattice-originated SHG and the magnetic-order-induced one (possibly both are electric dipole radiations), which has been reported in several papers already (see, for example, V. V. Pavlov et al., Phys. Solid State 45, 662 (2003) and S. Toyoda et. al., Phys. Rev. Mater. 7, 024403 (2023)). Importantly, these observations are not classified as (and not claimed to be) “optical nonreciprocity”, because electric dipole transition is an even function for the reversal of light momentum k . Previous works (refs. 12, 13) showing nonlinear optical diode effects utilize the interference between electric dipole SHG and magnetic dipole (or possibly quadrupole) one, with the latter changing sign upon the reversal of k .

At this stage, I’m not fully convinced that the observed phenomena can be referred to as “nonreciprocal” effect. If supposing that the sample is enough thin to have a transmission SHG signal, the nonreciprocity still survives?

We sincerely thank the reviewer for raising this important point. The reviewer insightfully pointed out that previous groundbreaking works (Refs. 13,14) on “nonreciprocal SHG” relied on magnetic dipole (or electric quadrupole) terms where the susceptibility contains contributions that are odd in the light momentum k . The reviewer therefore requested us to clarify the definition of our “nonlinear optical diode effect” in this context.

In short, the NODE is an effect whereby a nonlinear optical response (such as SHG), is stronger for one light propagation direction than for the reversed light propagation direction. Within this phenomenological picture, nonreciprocal SHG (NR-SHG) becomes a subset of the NODE. That subset indeed requires contributions to the susceptibility that are odd under k , as the reviewer insightfully pointed out. As we will show below, our NODE in general does not have such a requirement. Therefore, there is no contradiction. We entirely agree with the reviewer that it is very important to clarify our definition to avoid confusion to readers. To that end, we will also introduce below a microscopic foundation based on symmetry operations that will allow for a strict and meaningful

distinction between NR-SHG and NODE. We gratefully thank the reviewer for giving us the opportunity to explain our point of view in more detail.

1. Phenomenological definition of the nonlinear optical diode effect (NODE)

Before defining the NODE microscopically, we will consider the phenomenological definition given above whereby a nonlinear optical response (such as SHG), is stronger for one light propagation direction than for the reversed light propagation direction. We illustrate our definition of the NODE in the figure below (also Supplementary Fig. 4). In particular, in the context of optical experiments, it is intuitive to consider the “reversed effect” by swapping the light source and the detector. As such, we can define the NODE **both in a reflection and in a transmission geometry**. We can then define a directional contrast η in both geometries as

$$\eta = \frac{I^{\rightarrow}(2\omega) - I^{\leftarrow}(2\omega)}{I^{\rightarrow}(2\omega) + I^{\leftarrow}(2\omega)}$$

A contrast $\eta = \pm 1$ implies vanishing SHG intensity for one propagation direction (and finite intensity for the other), whereas $\eta = 0$ implies equal intensities for both propagation directions.

*Figure R4 Definition of the NODE. Phenomenologically, we define the nonlinear optical diode effect (NODE) as an effect whereby a nonlinear optical response (such as SHG), is stronger for one light propagation direction than for the reversed light propagation direction. Such an effect can be defined both in **a**, transmission and in **b**, reflection geometry. In both cases, we can define a directional contrast η . This figure is part of the Supplementary Information as Supplementary Figure 4.*

2. The previously observed nonreciprocal SHG

Within this definition, interestingly, the previous groundbreaking observations of the nonreciprocal SHG enable the transmission NODE. Because the incident and transmitted light beams are collinear, the transmission NODE necessarily requires the interference between one susceptibility even in k and one susceptibility odd in k , which is exactly the nonreciprocal SHG observed in Ref. 13 and 14 and insightfully pointed out by the reviewer.

The k -even and k -odd susceptibilities correspond to electric dipole SHG (ED-SHG) and magnetic dipole/electric quadrupole SHG (MD-SHG/EQ-SHG) in Refs. 13 and 14, respectively. Note that ED-SHG strictly requires a broken spatial inversion symmetry, whereas MD/EQ-SHG is allowed in centrosymmetric materials. In other words, ED-SHG is odd under spatial inversion \mathcal{P} , whereas MD/EQ-SHG is even under \mathcal{P} . Microscopically, NR-SHG is thus intricately linked to spatial inversion \mathcal{P} . The

presence of NR-SHG requires interference between \mathcal{P} -even and \mathcal{P} -odd contributions to the total SHG response. NR-SHG is thus only observable in systems with broken inversion symmetry.

Motivated by the microscopic definition of NR-SHG based on symmetry (here spatial inversion \mathcal{P}), we will now also define the NODE microscopically based on symmetry.

3. Microscopic definition of the NODE

To clearly distinguish NR-SHG and NODE, we need to define the NODE in terms of microscopic symmetry operations. Microscopically, the reversal of the propagation direction in Figure R4 is related to a mirror operation \mathcal{M} . The NODE arises if the total SHG response is comprised of contributions that are even and odd under that mirror operation \mathcal{M} .

$$\chi_{ijk} = \chi_{ijk}(\mathcal{M} \text{ even}) + \chi_{ijk}(\mathcal{M} \text{ odd}) \quad (1)$$

Interestingly, such a setting can be realized within the ED approximation (where all χ_{ijk} are odd with respect to \mathcal{P}). In the minimal example presented in Figure R5 below, the relevant mirror operation is \mathcal{M}_x (mirror plane perpendicular to x). The component χ_{xxz} is even under \mathcal{M}_x and χ_{xxx} is odd.

Figure R5 Minimal example for the observation and manipulation of the NODE. We consider a fictitious noncentrosymmetric material for which χ_{xxx} and χ_{xxz} are the only nonvanishing SHG tensor components in an orientation analogous to Fig.~1d in the main text. The detected SHG intensity is $I(2\omega) \sim |\sum_i P_i(2\omega)|^2$. Specifically for an angle of incidence of 45° , the electric field is parallel to **a**, $[1, 0, 1]$ in forward direction and **b**, parallel to $[1, 0, \bar{1}]$ in backward direction. Thus, $I^+(2\omega) \propto |\chi_{xxx} + \chi_{xxz}|^2$ and $I^-(2\omega) \propto |\chi_{xxx} - \chi_{xxz}|^2 \neq I^+(2\omega)$. We therefore observe a NODE in this minimal example. **c, d**, In a material where χ_{xxz} arises from a polar crystal structure whereas χ_{xxx} is due to magnetic order (as in the case of CeAlSi), switching the magnetization reverses the directionality of the NODE and thus allows its manipulation. This figure is also shown as Supplementary Fig. 5 in the Supplementary Information.

Given the observation of a directional contrast in this minimal example, which does not allow for NR-SHG (due to absence of \mathcal{P} -even SHG contributions) and the different microscopic origin (different relevant symmetries; \mathcal{P} vs. \mathcal{M}), we consider the NODE as a new optical interference phenomenon. The observation of a directional SHG response that is not due to NR-SHG is a novel distinction from the works of Mund et al. and Toyoda et al. with important implications for potential applications.

Note that the spatial inversion \mathcal{P} can be expressed as $\mathcal{P} = \mathcal{M}_x \circ \mathcal{M}_y \circ \mathcal{M}_z$. Therefore, a broken spatial inversion (as required for NR-SHG) implies a broken mirror symmetry, which enables a NODE. Thus, in addition to the observation that NR-SHG is in agreement with the phenomenological definition of the NODE in Figure R4, we also show formally based on microscopic symmetries that the

NODE is a proper generalization of NR-SHG. Note that we made no assumptions in the discussion above on whether χ_{xxz} or χ_{xxx} (or both or neither) are related to a magnetization in the material and thus their transformation behavior under time-reversal \mathcal{T} .

4. The role of time-reversal \mathcal{T}

We would like to point out that both NR-SHG and NODE are allowed in time-reversal symmetric (e.g., nonmagnetic) materials. However, a broken time-reversal symmetry \mathcal{T} further enriches the picture.

In particular, SHG tensor components can be even or odd with respect to \mathcal{T} (sometimes referred to as i-type and c-type, respectively), such that

$$\chi_{ijk} = \chi_{ijk}(\mathcal{T} \text{ even}) + \chi_{ijk}(\mathcal{T} \text{ odd})$$

Interference between such components can lead to a magnetic contrast that is frequently utilized to visualize magnetic domains. We can thus distinguish three optical interference phenomena based on the transformation behavior with respect to the symmetry operations \mathcal{P} , \mathcal{T} , or \mathcal{M} . The Venn diagram below summarizes the situation (also shown in the Supplementary Information as Supplementary Fig. 3).

Figure R6 SHG interference phenomena. Interference between different contributions to the total SHG response can give rise to a variety of optical phenomena. The phenomena can be distinguished based on the microscopic origin of the contributions. Magnetic contrast arises from a mixing of contributions that are even/odd under time reversal \mathcal{T} ; nonreciprocal SHG arises from mixing SHG contributions that are even/odd under spatial inversion \mathcal{P} ; the NODE emerges from interference between SHG contributions that are even/odd under a mirror operation \mathcal{M} . Previous demonstrations of NR-SHG are in the cross section of all three phenomena (Supplementary References 3 and 4 are Toyoda et al., *Sci Adv.* (2021) and Mund et al., *PRB* (2021), respectively). As a broken inversion symmetry (required for \mathcal{P} odd SHG) implies a broken mirror symmetry, any material that exhibits NR-SHG also allows for a NODE, i.e. NODE is a generalization of NR-SHG. This figure is also shown as Supplementary Fig. 3 in the Supplementary Information.

Interestingly, we find in the cross section of magnetic contrast and NODE all materials, where the mirror symmetry is broken by the magnetic order. This enables additional functionality beyond the bare demonstration of the NODE such as magnetic control of the directionality of the diode behavior. Our work lies squarely in the cross section.

Note that references 13 and 14 (Supplementary References 3 and 4; Toyoda et al., *Sci Adv.* (2021) and Mund et al., *PRB* (2021)) mentioned above consider the SHG response in a collinear setup in transmission geometry. However, nonreciprocal *linear* optical responses have been considered both in collinear transmission (e.g., Jung et al., *Phys. Rev. Lett.* 93, 037403 (2004)) and reflection geometries (e.g., Krichevstov et al., *J. Phys.: Condens. Matter* **5**, 8233 (1993)) as well as non-collinear reflection (e.g., Dumelow et al., *Phys. Rev. B* **54**, 12232 (1996)) and diffraction geometries (e.g., Kida et al, *Phys. Rev. Lett.* 96, 167202 (2006)). In analogy to these linear optical effects, we find it natural to also extend

directional nonlinear optical responses to noncollinear geometries. Specifically, we demonstrate in our work a NODE in a non-collinear reflection geometry. As the NODE in our work is based purely on electric-dipole-type SHG, it would be absent in transmission (assuming the CeAlSi sample would be thin enough to perform experiments in transmission).

It is definitely true that researchers have observed interference between multiple independent SHG tensor components (e.g., in the references mentioned by the reviewer). Such interference, however, is not in general dependent on the light propagation direction and can thus be distinguished from the NODE as we show in Supplementary Section 1D. Ultimately, one can distinguish between magnetic domain contrast, nonreciprocal SHG, and the NODE as three separate SHG interference phenomena. One important aspect of the novelty of our work therefore lies in the actual observation of a NODE in reflection by performing experiments in opposite directions.

5. Future experiments

Based on our systematic microscopic distinction between magnetic contrast, NODE, and NR-SHG, it would be interesting to find further examples of directional SHG responses. In particular, to the best of our knowledge, neither the NODE nor NR-SHG have been observed so far in time-reversal symmetric materials. Such observations, however, would significantly extend the number of materials available to realize devices utilizing the NODE.

In addition, we only consider here optical second-harmonic generation, but a nonlinear optical diode effect could arise also for other nonlinear optical responses, such as high-harmonic generation or photo-galvanic effects. The spectroscopic characterization of such effects may yield new insights into the underlying electronic structure of the materials.

We believe that our definition and first observation of the NODE opens up a new field that is fundamentally interesting and potentially technologically relevant.

6. Our revisions

In response to the points raised by the reviewer, we now clarify the definition of the NODE in the manuscript and its distinction from nonreciprocal SHG and magnetic SHG contrast (lines 60 – 74):

“We show in Fig. 1a a schematic representation of the NODE. From the point of view of symmetry, we note that the two opposite propagation directions are related by a mirror operation (mirror perpendicular to \hat{x} in Fig. 1a). The observation of a directionally dependent SHG intensity, i.e., a NODE, thus requires the presence of different contributions to the SHG intensity that are even and odd with respect to the mirror operation. This distinguishes the NODE from other SHG interference phenomena; most notably magnetic SHG contrast (Fiebig et al., JOSA B 22, 96 (2005)) and nonreciprocal SHG (Toyoda et al., Sci Adv. (2021) and Mund et al., PRB (2021)). Magnetic contrast arises due to interference between i-type (time reversal \mathcal{T} even) and c-type (\mathcal{T} odd) contributions to the SHG response and thus strictly requires \mathcal{T} symmetry to be broken. Nonreciprocal SHG arises from a \mathbf{k} -dependent nonlinear optical susceptibility (\mathbf{k} being the photon momentum). Therefore, nonreciprocal SHG contains \mathbf{k} -even contributions, such as electric-dipole SHG (which is \mathcal{P} odd) and \mathbf{k} -odd contributions, such as magnetic-dipole or electric-quadrupole SHG (which are \mathcal{P} even) (Toyoda et al., Sci Adv. (2021) and Mund et al., PRB (2021)). In contrast, the NODE is observable in the absence of \mathcal{T} or \mathcal{P} symmetry breaking or can be realized purely based on electric-dipole SHG. The NODE generalizes the ground-breaking concept of nonreciprocal SHG and may occur concomitantly with a magnetic contrast (see Supplementary Section 1.D for an extended discussion of SHG interference phenomena).”

Moreover, we recognize the potential of our work for future experiments in the main text (lines 273 – 278):

“in addition to searching for new diode effects based on other nonlinear optical processes, our results open up for a search for other materials exhibiting diode effects. So far, all observation of directional SHG responses were demonstrated in noncentrosymmetric, time-reversal-broken materials, although this should not strictly be necessary. It will be interesting to demonstrate the NODE in a time-reversal symmetric (e.g., antiferromagnetic) or inversion-symmetric systems.”

In addition, we now cite Pavlov et al., *Phys. Solid State* **45**, 662 (2003) and S. Toyoda et. al., *Phys. Rev. Mater.* **7**, 024403 (2023) as Refs. 43 and 44 in the main text.

We now further include an in-depth discussion of the various SHG interference phenomena in Supplementary Section 1.D. That discussion contains all arguments and definitions presented in this response.

To further avoid the impression of claiming optical nonreciprocity in our observations, we relabeled the “nonreciprocal contrast” to “directional contrast” (first introduced in line 197).

2.2. The broadband response is emphasized in the paper and discussed in relation to the linear dispersion of Weyl fermion in CeAlSi. However, generally, the interference of two electric dipole transitions in the SHG process can be broadband, as demonstrated in the references above.

Given that the broadband nature of the physical process discussed has been already reported in conventional oxides, meaning that the linear dispersion of Weyl electrons seems to be not necessary, the novelty of the present manuscript would not be so high.

We thank the reviewer for raising this important question about the role of the linearly dispersive bands for the observation of a broadband NODE. We fully agree with the reviewer that ED contributions to the SHG are often spectrally broader than MD-SHG contributions. Thus, by demonstrating a directional SHG response purely based on ED-SHG contributions, we already achieved a significant advancement in terms of available bandwidth over previous demonstrations of NR-SHG.

The spectral width of the ED-SHG is related to the band dispersion. We did not mean to claim that the actual presence of Weyl nodes is relevant. We agree that, even in the case of conventional semiconducting/insulating oxides, the dispersion of the valence band (for example) can broaden the optical response. We will show below that the bandwidth of the nonlinear optical response in CeAlSi is nevertheless exceptional. Based on density-functional theory, we relate this bandwidth to the electronic band structure of CeAlSi and identify strongly dispersive bands as the origin of the bandwidth. While such bands may also be present in other materials, they naturally occur in Weyl semimetals.

1. Broadband response

In terms of directional responses, we observe a NODE over the whole measured spectral range from 0.65 eV to 0.9 eV (bandwidth = 250 meV = 32% of the center energy). This NODE should be compared to other NODEs in literature. To our knowledge, the observations of nonreciprocal SHG by Toyoda et al. (Sci. Adv.) and Mund et al. (Refs. 13 and 14 of the main text) are the only other observations of a NODE albeit based on a different microscopic mechanism. The bandwidth in their observation is <10 meV (or <1% of the photon energy). We therefore believe that our observations can be referred to as a “broadband NODE”.

As the reviewer insightfully pointed out, interference of two ED-SHG contributions in general has been observed before. Many observations of magnetic contrast fall into this category including Pavlov et al., *Phys. Solid State* **45**, 662 (2003) and Toyoda et al., *Phys. Rev. Mater.* **7**, 024403 (2023). The NODE in CeAlSi is such that it also gives rise to a magnetic contrast (Figure R6). Although a discussion of the magnetic contrast is not the focus of our manuscript, we can compare the magnetic contrasts that are achievable by reversing the magnetization to previous observations of magnetic contrast.

In our work, we can achieve at least a six-fold change of SHG intensity by reversing the magnetization over the whole measured spectral range covering 0.65 eV to 0.9 eV (bandwidth = 250 meV = 32% of the center energy). Neither Pavlov et al. nor Toyoda et al. demonstrate such a strong contrast over any 250 meV range despite their higher photon energies. We are in fact not aware of any report of a strong magnetic SHG contrast that exhibits a similar bandwidth as our observations in CeAlSi (both absolute as well as relative to the photon energy). As a magnetic SHG contrast has not been observed before in topological semimetals, we investigate the role of the band structure for the extreme bandwidth in our observations.

Qualitatively, the linearly dispersive bands in CeAlSi allow for a wide range of photon energies to drive electronic transitions with comparable strength, which can explain the wide bandwidth for the SHG response. For a more quantitative argument, we turn towards density-functional theory.

2. Density-functional theory

Our numerically determined electronic band structure as well as the Weyl node positions (Supplementary Section 10A) are in agreement with other recent DFT calculations (see e.g., Yang et al., *Phys. Rev. B* **103**, 115143 (2021) and Sakhya et al., *Phys. Rev. Mater.* **7**, L051202 (2023)). We find a total of 40 Weyl nodes in the band structure of CeAlSi in the paramagnetic phase. The magnetic order does not change the number of Weyl nodes, but the position of the 40 Weyl nodes shifts both in energy and in momentum upon the inclusion of the magnetic order. Based on this electronic structure, we compute the spectrum of all SHG tensor components following the diagrammatic approach to nonlinear optical susceptibilities (Parker et al., *Phys. Rev. B* **99**, 045121 (2019); Takasan et al., *Phys. Rev. B* **104**, L161202 (2021)). We display the results in Figure R7 (also Supplementary Fig. 21). Strikingly, we notice an absence of sharp resonances.

Figure R7 First-principles calculations of SHG tensor component spectra in the magnetic phase of CeAlSi. All spectra are shown on the same scale. In agreement with the $2'1mm'$ point group symmetry, the grayed-out components vanish. The allowed components evolve continuously without any pronounced resonance in agreement with our understanding of broadband SHG due to linearly dispersive bands in CeAlSi. This figure is also shown as Supplementary Fig. 21 in the Supplementary Information.

The absence of resonances is in contrast to wide bandgap insulators, where often sharp resonances occur in the energy range close to the band gap. Based on the calculated SHG tensor components, we calculate the spectrum of the SHG intensity in a setting that reproduces the experiment in Fig. 1e of the main text (s-polarized incident light, p-polarized SHG response, 45° angle of incidence). We observe a clear difference in the SHG intensity between the two propagation directions. As a consequence of the absence of resonances in all tensor components, the SHG intensity evolves smoothly as a function of photon energy. Our numerical calculations thus reproduce a broadband SHG response.

Figure R8 Observation of the NOD from first principles. Spectral dependence of SHG intensity for counter-propagating light paths. The calculations are done based on the numerically obtained tensor components and reproducing the experimental setting of Fig. 1e in the main text. We observe a clear NOD. This figure is also shown as Supplementary Fig. 22 in the Supplementary Information.

3. Relation to linearly dispersive bands

Numerically, the nonlinear optical susceptibilities χ_{ijk} are expressed as k-space integrals in the form $\chi_{ijk} = \int_{BZ} \xi_{ijk} d^3\mathbf{k}$. To gain further insights into the relation of the SHG response and the electronic structure of CeAlSi, we investigate the k-space resolved origin of the SHG response by studying ξ_{ijk} . As an example, we focus here in particular on ξ_{xxy} in the ferromagnetic M_{+x} state (magnetization pointing along [110]).

In Figure R9, we display the magnitude of ξ_{xxy} at an incident photon energy of 650 meV. We choose here a plane in the Brillouin zone with $k_z = 0.295 \text{ \AA}^{-1}$. In the paramagnetic phase of CeAlSi, this plane contains the W_2 Weyl nodes. In the ferromagnetic phase, the Weyl nodes shift out of the plane. Red and blue dots in Figure R9 therefore indicate the projection of the Weyl nodes onto the plane $k_z = 0.295 \text{ \AA}^{-1}$ in the ferromagnetic phase.

Figure R9 k space distribution of ξ_{xxy} from first principles. **a**, k space distribution of ξ_{xxy} in the plane $k_z = 0.295 \text{ \AA}^{-1}$ for an incident photon energy of 650 meV. Red and blue dots indicate the projection of the W_2 Weyl nodes of different chirality onto the considered plane. **b**, contributions to ξ_{xxy} from electronic transitions between electronic bands -1 and +1. **c-g**, Contributions to ξ_{xxy} at various incident photon energies between 550 meV and 950 meV. Only the section within the dashed box of panel b is shown. **h**, schematic band structure illustrating the origin of the two bands surrounding the Weyl node pair in panels c-g: the closer band corresponds to an electronic transition at the incident photon energy $\hbar\omega$, whereas the outer band corresponds to an electronic transition at $2\hbar\omega$. This figure is also shown as Supplementary Fig. 23 in the Supplementary Information.

We notice a rather complicated distribution of $|\xi_{xxy}|$ in the plane, but major contributions appear to be related to the Weyl nodes. Note that ξ_{xxy} contains a sum over all bands. The main features of Figure R9a, however, can already be recognized in Figure R9b, where we only consider contributions from electronic transitions between bands -1 and +1 across the Fermi level (marked red and blue in Fig. 4c of the main text).

Note that each pair of Weyl nodes is surrounded by two lines of strong contributions to ξ_{xxy} . The inner line corresponds to electronic transitions that are resonant at the incident photon energy (here: 650 meV); the outer line corresponds to transitions resonant at the second-harmonic photon energy (here: 2x 650 meV).

As we vary the incident photon energy between 550 meV (Figure R9c) and 950 meV (Figure R9g), we do not observe major changes, but the distance of the inner and outer lines to the Weyl node pair changes. In fact, the distance of the lines of ξ_{xxy} contributions from the Weyl nodes increases approximately linearly with the incident photon energy (dashed black lines in panels c-g), which is due to the linearly dispersive bands in the vicinity of the Weyl nodes.

In Figure R9h, we show a schematic illustration of a possible band structure near the line with $k_{[010]} = 0$, which would qualitatively reproduce our observations from the DFT calculations.

Interestingly, the magnitude of ξ_{xxy} does not change significantly within the considered energy range (in line with the smooth changes of χ_{ijk} , Figure R7). The comparable magnitude of ξ over a wide energy range in combination with smooth changes of the ξ contributions throughout the Brillouin zone rationalize the broadband SHG response that we observed in the experiment and establish a connection between the response and linearly dispersive bands in CeAlSi.

Note that not only the electronic bands containing the Weyl nodes are linearly dispersive. In fact, we can find in most bands near the Fermi energy sections that are linearly dispersive over a range of hundreds of meV (Supplementary Fig. 15), which enable the nonlinear optical broadband response. In CeAlSi, these sections typically occur in the vicinity of the Weyl nodes. While the actual presence of the Weyl nodes may not be strictly necessary, they naturally give rise to linearly dispersive bands making Weyl semimetals prime candidates for the observation of a broadband NODE.

In terms of novelty, our measurements represent the first observation of magnetic SHG contrast in any topological semimetal. We further introduce the NODE as a novel concept that generalizes nonreciprocal SHG. We report the observation of a strong NODE over a wide spectral range – even in comparison to other, non-directional, SHG interference phenomena. This is also the first observation of a NODE in reflection. Supported by density-functional theory, we attribute the broadband response to linearly dispersive bands in CeAlSi. Such a relation of experimentally observed optical responses to specific features of the electronic band structure is rarely achieved.

4. Our revisions

We now include the panels of Figure R9d and f in Fig. 4 (panels f and g) in the main text and discuss the role of the dispersive bands more carefully in the main text (lines 234 – 245): “We further elucidate the k -space origin of the broadband SHG response in Fig. 4f and g. We consider as an example the distribution of $|\xi_{xxy}|$ in the plane $k_z = 0.295 \text{ \AA}^{-1}$. In the paramagnetic phase of CeAlSi, this plane contains the W_2 Weyl nodes. In the ferromagnetic phase, the Weyl nodes shift out of the plane. Red and blue dots therefore indicate the projection of the Weyl nodes onto the plane $k_z = 0.295 \text{ \AA}^{-1}$. The pair of Weyl nodes in Fig. 4f are enveloped by two lines of strong contributions to $|\xi_{xxy}|$. In analogy to Fig. 4d, we identify the inner and outer lines as transitions that are resonant at $\hbar\omega$ and

$2\hbar\omega$, respectively. As we increase the incident photon energy from 650 meV (Fig. 4f) to 850 meV (Fig. 4g), we observe that the lines of strong contributions occur at an increased distance from the Weyl nodes (as indicated by the dashed line). This observation is a direct consequence of the linearly dispersive bands that disperse over hundreds of meV in CeAlSi. We thus see that the broadband NODE in CeAlSi ultimately arises from the linearly dispersive bands that naturally occur in the vicinity of the Weyl nodes (see also Supplementary Section 10 for an extended discussion).”

As we included new panels to Fig. 4, we also extended the figure caption: “**f, g, Distribution of $|\xi_{xxy}|$ in a section of the Brillouin zone plane $k_z = 0.295 \text{ \AA}^{-1}$ for $\hbar\omega = 650 \text{ meV}$ and 850 meV , respectively. Red and blue dots indicate the projection of the W_2 Weyl nodes of different chirality onto the considered plane. As the photon energy increases, contributions to $|\xi_{xxy}|$ occur at larger distance from the Weyl nodes -- a direct consequence of linearly dispersive bands. Dashed line is a guide to the eye highlighting the expansion of the contribution line with increased photon energy.**”

In addition, we now extended our numerical analysis and provide important further details on the numerical calculations in Section 10 of the Supplementary Information. We first introduce the electronic structure of CeAlSi (Section 10.A). We further provide an in-depth analysis of the k-space and band resolved contributions to the SHG response in Section 10.B, where we provide further evidence for the role of linearly dispersive bands in CeAlSi for the nonlinear optical response. The discussions in the newly added Supplementary Information sections contain all arguments presented in this response.

2.3. In Fig. 1, please compare the experimental SHG spectra and DFT one directly. This is necessary to confirm the quality of these data.

We thank the reviewer for this suggestion. We agree that our combined theoretical and experimental work opens up for the opportunity to compare measurement and simulation.

In general, there are only very few reports of combined experimental and theoretical works showing numerically calculated magnetic SHG tensor components (e.g., Toyoda et al., *Phys. Rev. Mater.* **7**, 024403 (2023)). Even in those works, a quantitative comparison between experiment and theory is often not feasible as nonlinear optical responses are notoriously difficult to calculate from first principles. Nevertheless, this is less an indication of low data quality and more testament to the complexity of the experiment. The measured SHG spectra are affected not only by the spectral dependence of the nonlinear optical susceptibility, but also by the refractive index and absorption at both the incident fundamental and outgoing second-harmonic photon energy. In addition, surface roughness or adsorbates may further distort the spectrum. Although we took great care to maintain a pristine and clean sample surface (e.g., sample storage in vacuum, all measurements done on naturally cleaved surface), this adds further uncertainty.

Nevertheless, we can compare our data and density functional calculations qualitatively. The numerically determined spectra for all tensor components (Figure R7; Supplementary Fig. 21) evolve smoothly as a function of photon energy and do not exhibit any discernable resonances. The absence of any strong resonances is in agreement with our data. In addition, and following the suggestion of the reviewer, we now show in Figure R8 (and Supplementary Fig. 22) a calculated spectrum of the SHG intensity. In agreement with the experiment, we do not observe any sharp features. Moreover, we observe a significant difference in the calculated SHG intensity for the two propagation directions. Note that our calculations only consider electric-dipole-type SHG. This is numerical proof that a NODE can be observed within the electric-dipole approximation. Based on the general, qualitative

agreement between our measurements and simulations, we are confident that the observed NODE is an intrinsic property of CeAlSi.

In addition, we can judge the experimental quality of the data by considering the reproducibility. For example, we can compare the NODE in M_{+y} state (Fig. 1e in the main text) to the NODE in the M_{-y} state (Supplementary Fig. 9; both plots shown in Figure R10 below for your convenience). The spectra for different propagation directions were recorded several days apart. In between, the sample was brought to room temperature, the cryostat was moved, the experimental setup was rebuilt for the opposite propagation direction, and the laser hit the sample in a slightly different spot. Nevertheless, the symmetry-related spectra $I^{\rightarrow}(+y)$ and $I^{\leftarrow}(-y)$ exhibit all the same features at the same wavelengths (similarly $I^{\rightarrow}(-y)$ and $I^{\leftarrow}(+y)$). The same qualitative and quantitative reproducibility can be found by comparing symmetry-related panels in Supplementary Fig. 10 or Supplementary Fig. 11. Overall, the high degree of reproducibility indicates a high quality of our data.

Figure R10 Data reproducibility. As an example, we compare **a**, the NODE in M_{+y} state (Fig. 1e) to **b**, the NODE in the M_{-y} state (Supplementary Fig. 9). The symmetry-related spectra $I^{\rightarrow}(+y)$ and $I^{\leftarrow}(-y)$ exhibit all the same features at the same wavelengths (similarly $I^{\rightarrow}(-y)$ and $I^{\leftarrow}(+y)$) indicating a high quality of our data.

2.4. Fig. 2: Please show the incident angle dependence. It should be possible to experimentally deduce the amplitude of each tensor elements, to understand the details of interference.

We thank the reviewer for this suggestion. We agree that such an analysis should be in principle possible.

It is indeed possible to gain some insight by utilizing our polarization resolved SHG experiments in forward and backward propagation direction, which corresponds to measurements at an angle of incidence (AoI) of -45 deg and $+45$ deg, respectively (Supplementary Fig. 6). In fact, the sixteen polarization dependences below T_c (4 magnetization directions, 2 analyzed SHG polarizations s and p , 2 propagation directions) were all fit simultaneously with the same nonlinear optical susceptibility based on the allowed ED-SHG components in the magnetic point group $2'_{mm'}$. As a result of the fitting procedure, we now provide the relative magnitudes and phases of all allowed SHG tensor components in Supplementary Section 2.

From the fit results, we can numerically determine the incident angle dependence for any polarization of incident and outgoing laser beams. We show in Figure R11 (also Supplementary Fig. 7) the calculated incident angle dependence for s -polarized incident light, p -polarized outgoing SHG in the

magnetic M_{+y} state. The inset shows the calculated absolute value of the directional contrast, i.e., the contrast between the reflection at some angle of incidence (AoI) and the opposite incidence angle ($-AoI$). The directional contrast is thus defined for pairs of incident angles (AoI, $-AoI$). Although we observed a strong directional contrast at an incident angle of $|45^\circ|$, this angle is in general not optimal. Unfortunately, it is not feasible in our setup to continuously vary the incidence angle in a reflection geometry.

Figure R11 Incident angle dependence. Calculated incident angle dependence based on the relative magnitudes and phases of the SHG tensor component extracted from experimental polarization dependencies. An angle of incidence (AoI) of 45° is not optimal. By adjusting the angle of incidence, a numerically determined directional contrast $>99.9\%$ could be possible (inset). Black circles mark experimental configurations in our work ($\pm 45^\circ$). This figure is also shown as Supplementary Fig. 7 in the Supplementary Information.

We have shown that a NODE can occur at oblique angles of incidence. The directional contrast quantifying the NODE must be a continuous function of the angle of incidence. It is therefore interesting to consider two edge cases:

1. Normal incidence (AoI = 0°)

In this scenario, incident and outgoing beams overlap. A reversal of the light path (AoI \rightarrow $-AoI$) does not change the experiment. Hence, a NODE cannot be observed in normal incidence reflection.

2. Grazing incidence (AoI = 90°)

Also here, a NODE cannot be observed in the electric-dipole approximation. In the coordinate system of Figure R5, grazing incidence corresponds to light propagation parallel to \mathbf{x} . The transversal electric fields thus only couple to tensor components χ_{ijk} with $i, j, k \in \{y, z\}$, which are all even with respect to the mirror operation \mathcal{M}_x . Hence, the fundamental symmetry requirement for the observation of a NODE (Equation (1)) cannot be realized.

Our revisions:

We now provide the relative magnitudes and phases of all allowed SHG tensor components in Supplementary Section 2. There, we also include the interesting discussion about the angle of incidence, which highlights a promising control parameter for the optimization of the NODE in future works and applications.

2.5. In the movie of current-induced control, the domain structures change largely even outside of the current path. Please elaborate this point. I think, discussions (on the effective magnetic field etc.) should be given in the main text, not in the figure caption.

We are grateful to the reviewer for this nice question. The reviewer is absolutely right. The current-induced magnetization switching affects the magnetic domains even outside of the electrode pattern.

We attribute this observation to a pronounced magneto-elastic coupling in CeAlSi [Xu et al., *Adv. Quantum Technol.* **4**, 2000101 (2021)]. The magnetic state is coupled to strain, which in turn propagates beyond the electrode pattern and thus changes the domain distribution outside of the electrode area.

We further observed in our spatially resolved domain images a strong tendency of the magnetic domain walls to align with x and y directions as well as their bisectors (Figure R12, also part of Supplementary Fig. 13). Due to this behavior, which is itself likely rooted in strain and magnetic stray field minimization, domain walls cannot easily be tilted. Thus, changes of the domain distribution tend to propagate beyond the originally affected area, such as the electrode area in case of current-induced control.

Figure R12 **Domain Imaging**. Spatially resolved domain image of CeAlSi in the absence of magnetic field and current. Domain walls align with x and y directions as well as their bisectors. This figure is part of Supplementary Fig. 13.

We list this strain effect together with other possible mechanisms that can give rise to an effective magnetic field in lines 148 – 152 of the main text:

“The noncentrosymmetric symmetry of CeAlSi supports a magnetoelectric coupling between the magnetization \mathbf{M} and a current \mathbf{J} . Microscopically, such a coupling may be either intrinsic^{34,35}, mediated by strain³⁶, or relying on the spin-Hall effect³⁷ and assisted by the Oersted field of the current^{38,39}. In all cases, the magnetoelectric coupling is such that a current in the xy-plane favors a magnetization perpendicular to the current in the xy-plane with a fixed handedness...”

We further agree with the reviewer that the remark on effective magnetic fields in the figure caption is more confusing than helpful. We removed the remark.

Minor:

2.6. Experimental conditions are not fully described. For example, it is not clear that at which temperature the experiments are executed.

Thank you for flagging this issue. Now we specify all experimental conditions in the methods section and in the Supplementary Information. In particular, all measurements shown in the main text were performed at a temperature of 3 K.

We write in the methods section (lines 408 – 409):

“All measurements were performed at 3 K unless explicitly stated differently. In particular, all measurements shown in the main text were performed at 3 K.”

2.7. Fig. 1 caption: Please define what is “localized electronic transition” means here.

We were referring to transitions between localized electronic states, i.e., states that are energetically flat (non-dispersive) throughout the Brillouin zone. The picture applies for example to $4f$ states or $d-d$ transitions in wide-gap semiconductors. We recognize that the term “localized transition” is not widely used.

We now write “localized states” in Fig. 1 and specify explicitly “transition between localized electronic states” in the figure caption.

2.8. As for the experiments on the microdevice (Fig. 3h and Extended data Fig. 7), please explain which tensor elements are active, to analyze the observed anisotropy pattern.

We thank the reviewer for giving us the opportunity to explain our experiments on the microdevice in more detail. In short, all tensor components that are allowed by the $2'mm'$ symmetry are active.

Although originally the lines in Fig. 3h and former Extended data Fig. 7 were only guides to the eye and not fit results, we now analyze the SHG polarization dependence for the different magnetic states of the microdevice in detail in Supplementary Section 9. Solid lines in Supplementary Fig. 17 (also shown as Fig. R13 below) are now fit results. The values of the fit parameters are given in Supplementary Table V. Differences in the tensor components between the fit results in Supplementary Table V and Supplementary Table IV (for the bulk single crystal) are likely related to slight changes of the sample surface due to the focused ion beam milling in the preparation of the microdevice. Nevertheless, we robustly observe a NODE even on the microdevice.

Figure R13 **SHG Polarization dependencies for FIB sample**. Analogously to the SHG signal from CeAlSi bulk single crystals (Supplementary Fig. 6), the SHG polarization dependence allows us to distinguish the four magnetization directions in micro-machined CeAlSi samples. Here, the magnetic state was stabilized by an external magnetic field of 30 mT. Solid lines are fits. A comparison of field-polarized measurements to the field-free measurements shown in Figs. 3h,i reveals electrical switching between M_{-x} and M_{+x} states, respectively. This figure is also shown as Supplementary Fig. 17 in the Supplementary Information.

In addition to the added **Supplementary Section 9** in the Supplementary information, we now **updated panels h and i of Fig. 3** (also shown here as Figure R14 for your convenience). Figures 3h and i now contain fits corresponding to the magnetic M_{-x} and M_{+x} state, respectively. We adjusted

the figure caption accordingly: “**h,i**, SHG polarization dependence under application of -3 mA and +3 mA, respectively. Changes are most prominent for s-polarized SHG (red). Solid lines are fits corresponding to magnetic M_x and M_{+x} states, respectively (Supplementary Section 9).”

Figure R14 **Electrical control of the NODE.** **h,i**, SHG polarization dependence under application of -3 mA and +3 mA, respectively. Changes are most prominent for s-polarized SHG (red). **Solid lines are fits corresponding to magnetic M_x and M_{+x} states, respectively (Supplementary Section 9).** These panels are part of Figure 3 in the main text.

REVIEWERS' COMMENTS

Reviewer #1 (Remarks to the Author):

The authors have revised and improved their manuscript according to reviewers' questions and suggestions. I recommend the publication.

Reviewer #2 (Remarks to the Author):

I found that the replies from the authors are thorough and reasonable. Therefore, now I would like to recommend this paper to be published in Nature Communications.